# ReplaceMe: Network Simplification via Depth Pruning and Transformer Block Linearization

**Dmitriy Shopkhoev** *
MWS AI, ITMO University

**Ammar Ali** *
MWS AI, ITMO University

**Magauiya Zhussip**
MWS AI

**Valentin Malykh**
MWS AI, ITMO University,
IITU University

**Stamatios Lefkimmiatis**
MWS AI

**Nikos Komodakis**
University of Crete, IACM-Forth,
Archimedes Athena RC

**Sergey Zagoruyko**
Polynome

## Abstract

We introduce **ReplaceMe**, a generalized training-free depth pruning method that effectively replaces transformer blocks with a linear operation, while maintaining high performance for low compression ratios. In contrast to conventional pruning approaches that require additional training or fine-tuning, our approach requires only a small calibration dataset that is used to estimate a linear transformation, which approximates the pruned blocks. The estimated linear mapping can be seamlessly merged with the remaining transformer blocks, eliminating the need for any additional network parameters. Our experiments show that ReplaceMe consistently outperforms other training-free approaches and remains highly competitive with state-of-the-art pruning methods that involve extensive retraining/fine-tuning and architectural modifications. Applied to several large language models (LLMs), ReplaceMe achieves up to 25% pruning while retaining approximately 90% of the original model's performance on open benchmarks—without any training or healing steps, resulting in minimal computational overhead. We provide an open-source library implementing ReplaceMe alongside several state-of-the-art depth pruning techniques, available at https://github.com/mts-ai/ReplaceMe.

## 1 Introduction

In recent years, transformers have achieved unprecedented success across a wide range of tasks on both computer vision and natural language processing. Modern large language models (LLMs) typically scale up to billions or even hundreds of billions of parameters, significantly increasing the computational and memory requirements for both training and inference stages. This substantial resource demand poses a critical challenge for their wider practical deployment and usability.

Due to the excessive size of modern LLMs, there has been significant research effort to make such models accessible to users with limited hardware capabilities. These efforts primarily focus on three key strategies: quantization, distillation, and pruning. Pruning, which is the focus of the current work, involves identifying and removing less important parameters or entire structural components to streamline the model, thereby reducing computational overhead without significantly compromising the performance. Structured pruning is different from unstructured pruning in that it focuses on entire groups of parameters or layers, allowing their complete removal. This approach not only

---

*Equal contribution

39th Conference on Neural Information Processing Systems (NeurIPS 2025).

enhances hardware utilization efficiency but also potentially achieves greater reductions in resource consumption. Importantly, it operates independently of the hardware type used.

In this work, we focus on structural depth pruning, operating under the hypothesis that a contiguous set of transformer blocks can be effectively approximated by a single linear transformation. To validate this idea, we propose ReplaceMe, a novel training-free pruning method that replaces selected blocks with a linear transformation estimated from a small calibration dataset. It should be noted that most existing pruning methods require a post-pruning retraining phase, often referred to as a "healing process", to recover lost performance. This retraining stage can be time-consuming and computationally expensive. In contrast, ReplaceMe preserves the majority of the model performance *without any retraining* for reasonable compression ratio scenarios. ReplaceMe generalizes depth pruning methods by introducing a simple yet effective linear transformation that compensates for the error caused by block removal. This transformation is subsequently fused with one of the remaining model weights, enabling seamless integration without adding parameters. The contributions of this work can be summarized as follows:

1. We propose ReplaceMe, a generalized method for depth pruning that can maintain model performance without requiring any healing process, for reasonable compression ratios;

2. We conduct a detailed study on the estimation of the linear transformation with both analytical and numerical methods and under different objectives;

3. We provide detailed ablation studies for different calibration data, solvers, and LLM architectures;

4. We validate the effectiveness and generality of ReplaceMe across diverse model families, including LLMs and vision transformer architectures like ViT [7].

This paper is organized as follows: Section 2 presents the core methodology behind our training-free depth pruning approach. It introduces the framework for identifying prunable layers in large language models (LLMs) and estimating the corresponding linear transformations that compensate for the removed components. This section also discusses the selection of appropriate loss functions, regularization strategies to ensure generalizability, and the potential extension to multiple linear transformations for more flexible pruning. Section 3 then provides comprehensive experimental results and ablation studies, demonstrating the effectiveness and robustness of our method, and analyzing the key factors that influence its performance.

## 2 Method

Next, we introduce ReplaceMe, a novel depth-wise neural network pruning method that balances simplicity and effectiveness to optimize model performance. Our approach is based on the idea of pruning multiple layers in transformer models and replacing them with a single linear transformation. It consists of the following key steps: First, we identify layers suitable for pruning by targeting those with minimal impact on performance, in line with prior research (Section 2.1). Next, we compute an optimal linear transformation (LT) to compensate for the contributions of the pruned layers. Notably, this transformation is seamlessly integrated into the preceding layer, preserving model performance without introducing additional parameters (Section 2.2). Furthermore, we study the effect of regularization methods on the estimation of the transformation and show that this can be a helpful strategy to maintain balance between model performance and perplexity (section 2.3). Finally, we outline how to extend our framework to support multiple linear transformations, enabling flexible and informed pruning decisions (section 2.4). Together, these components establish ReplaceMe as a practical and robust advancement in training-free neural network pruning.

### 2.1 Layers selection

Let $\mathbf{X}_i \in \mathbb{R}^{N \times d}$ be the input to the $i$-th transformer block, where $N$ denotes the number of tokens and $d$ the hidden dimension of the transformer model. Then, typically, the conventional transformer

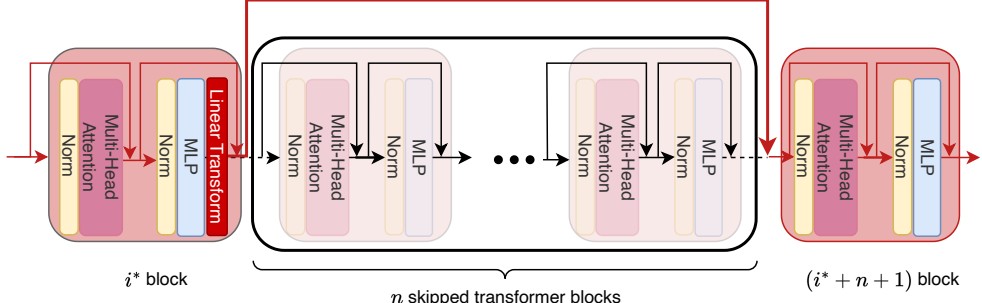

Figure 1: ReplaceMe compresses and accelerates LLMs by bypassing a contiguous sequence of transformer blocks—illustrated by the red line—while preserving model performance. This is achieved by inserting an estimated linear transformation matrix that maps the MLP output of the $i^*$-th block directly to the input space expected by the $(i^* + n + 1)$-th block, effectively replacing all $n$ blocks in between.

block can be expressed in the following way:

$$\mathbf{Y}_i = \mathbf{X}_i + \mathrm{MHA}_i\big(\mathrm{LN}_i^{(1)}(\mathbf{X}_i)\big) \tag{1}$$

$$\mathbf{M}_i = \mathrm{MLP}_i\big(\mathrm{LN}_i^{(2)}(\mathbf{Y}_i)\big) \tag{2}$$

$$\mathbf{L}_i = \mathbf{Y}_i + \mathbf{M}_i, \tag{3}$$

where $\mathrm{MHA}_i$ and $\mathrm{MLP}_i$ denote the multi-head attention (MHA) and MLP layers, respectively, while $\mathrm{LN}_i^{(1)}, \mathrm{LN}_i^{(2)}$ correspond to the layer normalization operators before the MHA layer and after the MLP, respectively. The output of the attention sub-block is denoted as $\mathbf{Y}_i$, while $\mathbf{M}_i$ and $\mathbf{L}_i$ represent the output of the MLP layer and the transformer block, respectively.

The layer selection strategy is based on the significance of each layer, which is determined by the distance between the activation outputs of different transformer blocks. Formally, for a predefined number of layers to be pruned, denoted as $n$, the optimal cut index $i^*$ is determined by minimizing the distance between hidden states before and after the cut:

$$i^* = \arg\min_i h\left(\mathbf{L}_i, \mathbf{L}_{i+n}\right). \tag{4}$$

We evaluated various distance metrics $h(\cdot)$ using activations computed on a small calibration dataset, with the impact of the dataset explored in Section 3.3.1. For each candidate cut, the distance between hidden states before and after the cut is calculated and then averaged across the calibration dataset to obtain a robust layer-importance estimate. We found that the cosine distance is particularly effective in identifying nearly optimal layers for pruning. This observation aligns with findings reported in recent studies [13]. In the supplementary material A.10, we present results of an exhaustive brute-force layer selection, which confirm that cosine distance consistently identifies optimal or near-optimal layers for removal. Additionally, we provide a comparative analysis of the $L_2$ distance metric to further validate our choice.

## 2.2 Linear Transform Estimation

To compensate for the pruned transformer blocks, we leverage the same set of calibration data to compute activations directly before and after the removal point. Using these activations, we estimate an appropriate linear transformation that accurately approximates the final output of the pruned blocks. Depending on the selected criterion, this estimation can be performed using either analytical or numerical approaches, enabling precise modeling of the omitted layers.

As illustrated in Fig. 1, a linear transformation is applied following the MLP and prior to the residual summation. The goal of our method is to estimate an optimal linear transformation matrix $\mathbf{T}$ so that:

$$\mathbf{T}^* = \arg\min_{\mathbf{T}} h\big(\mathbf{M}_i \cdot \mathbf{T} + \mathbf{Y}_i; \mathbf{L}_{i+n}\big) \tag{5}$$

where $h(\cdot)$ corresponds to a distance function (e.g., $L_2$ distance, cosine distance, etc.) between two input tensors and $i$ denotes the optical cut index estimated using Eq. (4). Once the transformation matrix is estimated, the transformer blocks from $i+1$ to $i+n$ (inclusive) are removed.

$L_2$**-Distance.** A classical solution for Eq. 5 arises when setting $h(\cdot)$ as the $L_2$-distance and is obtained by solving a least-squares (LS) problem:

$$\mathbf{T}^* = \arg\min_{\mathbf{T}} \left\| (\mathbf{M}_i \cdot \mathbf{T} + \mathbf{Y}_i) - \mathbf{L}_{i+n} \right\|_2^2 = (\mathbf{M}_i^T \cdot \mathbf{M}_i)^{-1} \cdot \mathbf{M}_i^T \cdot \left( \mathbf{L}_{i+n} - \mathbf{Y}_i \right). \qquad (6)$$

For a detailed derivation of the above result we refer to the supplementary material A.2.

**Cosine Distance.** As discussed in Section 2.1, our ablation study on various distance functions to assess the importance of transformer blocks, revealed that the cosine distance is the most effective in identifying the least significant blocks. Motivated by these results, we have further used the cosine distance as the objective function to estimate the optimal linear transformation. In this case, the optimization problem takes the form:

$$\mathbf{T}^* = \arg\min_{\mathbf{T}} \cos(\mathbf{M}_i \cdot \mathbf{T} + \mathbf{Y}_i, \mathbf{L}_{i+n})$$

$$= \arg\min_{\mathbf{T}} \sum_{k=1}^{N} \left( 1 - \frac{(\mathbf{M}_{i,k} \cdot \mathbf{T} + \mathbf{Y}_{i,k})^{\mathsf{T}} \cdot \mathbf{L}_{i+n,k}}{\|\mathbf{M}_{i,k} \cdot \mathbf{T} + \mathbf{Y}_{i,k}\|_2 \|\mathbf{L}_{i+n,k}\|_2} \right), \qquad (7)$$

where $\cos(\cdot, \cdot)$ denotes the cosine distance between the two input vectors. We use the notation $\mathbf{M}_{i,k} \in \mathbb{R}^d$ to denote the $k$-th row of a matrix $\mathbf{M}_i \in \mathbb{R}^{N \times d}$, which we then represent as a column vector. Here, the cosine distance is calculated per token $k$ and aggregated over all $N$ tokens. Unlike the $L_2$-distance formulation, this objective does not admit a closed-form solution, and thus a numerical optimization approach is required. In our experiments, we have utilized the Adam [19] optimization algorithm. Furthermore, an ablation study involving various alternative numerical solvers can be found in the supplementary material A.6.

To solve the optimization problem in Eq.(7), it would be necessary to store the hidden states $\mathbf{M}_i$, $\mathbf{Y}_i$, and $\mathbf{L}_{i+n}$. To improve memory efficiency, we instead optimize the following simplified formulation:

$$\mathbf{T}^* = \arg\min_{\mathbf{T}} \cos(\mathbf{M}_i \cdot \mathbf{T}, \mathbf{L}_{i+n} - \mathbf{Y}_i). \qquad (8)$$

This alternative formulation requires us to store only $\mathbf{M}_i$ and the difference $\mathbf{L}_{i+n} - \mathbf{Y}_i$, instead of keeping all three matrices. In supplementary material A.11, we empirically demonstrate that this simplification has a negligible effect on performance while improving memory efficiency.

**Fusing the Linear Transformation** Once the optimal transformation $\mathbf{T}^*$ has been estimated, our approach allows it to be incorporated into the MLP layer of the $i$-th transformer block. Since the learned transformation and the pre-cut MLP block represent a sequential linear operators, they can be algebraically composed into a single equivalent linear transformation, allowing $\mathbf{T}^*$ to be fused with the weight matrix of the down projection linear layer of the MLP block. Consequently, the overall architecture of the model remains unchanged, except for the removal of the "non-effective" transformer blocks.

## 2.3 Regularization

We further consider estimating the linear transformation by imposing additional constraints on the matrix $\mathbf{T}$ through regularization. Specifically, we reformulate the optimization problem as follows:

$$\mathbf{T}^* = \arg\min_{\mathbf{T}} h(\mathbf{M}_i \cdot \mathbf{T} + \mathbf{Y}_i; \mathbf{L}_{i+n}) + \alpha \cdot R(\mathbf{T}), \qquad (9)$$

where $R(\cdot)$ denotes the regularizer and $\alpha$ controls its strength. To promote sparsity in the transformation matrix $\mathbf{T}$ and encourage a more balanced distribution of feature importance, we use both $L_1$ and $L_2$ regularization terms when we use the cosine distance as our objective. When we instead utilize $L_2$ as our objective, we derive the analytical solution under $L_2$ regularization. Empirical analysis shows that the considered regularization approaches improve the ability of the pruned model to generate accurate predictions, as reflected by the accuracy-based benchmarks (see Sec. 3). Though, such improvement comes at the cost of increased perplexity.

## 2.4 Multiple Linear Transforms

The proposed ReplaceMe method can be easily extended to be applied on multiple non-overlapping groups of blocks within the model, estimating a separate linear transformation for each group (multi-LT). This approach provides flexibility in achieving the desired performance metrics, even under high compression ratios. Furthermore, if some of the selected groups of blocks are consecutive, they can be merged into a single block with one corresponding linear transformation. We refer to this method as non-consecutive Multi-linear transformations (Multi_LT_NC). Experimental analyses further validating Multi_LT_NC are detailed in the Appendix A.16.

# 3 Experiments

In this section, we first describe our experimental setup and then provide a systematic comparison of our training-free pruning method against existing structured pruning approaches, including those relying on healing mechanisms. In particular, we show that our method achieves competitive performance without requiring additional training. To further analyze the factors influencing our approach, we conduct ablation studies on several key aspects: the calibration dataset (used for choosing the layers to be pruned and for estimating the linear transformation), the choice of distance function in Eq. 5, and the impact of regularization in the linear transform estimation.

## 3.1 Experimental setup

In our experiments, we have focused primarily on LLaMA-2-7B and LLaMA-3-8B-Instruct models, while also reporting results for Qwen2.5-7B and Falcon-11B for a comparative analysis. Results for additional models are provided in the supplementary material A.7. For numerical estimation of the linear transform, we used Adam optimizer with LR $1e^{-4}$ and batch size 1024, iterating for 10 epochs over the calibration data. In Table 1 we present results on different benchmarks that have been widely used in previous research. These benchmarks have been introduced in the following works: CMNLI [56], HellaSwag [57], PIQA [2], CHID [59], WSC [24], MMLU [16], CMMLU [25], Race-High/Middle [22], C3 [46]. Additionally, we benchmarked ReplaceMe using well-established public datasets, namely Winogrande [41], BoolQ [4], OpenBookQA [31], SciQ [55], and Lambada OpenAI [35]. For all benchmarks except Lambada OpenAI, we report accuracy as the evaluation metric, along with the average accuracy across all benchmarks. For Lambada OpenAI, we report perplexity.To calculate the environmental impact we used CodeCarbon [6], a well-known tool to estimate $CO_2$ emissions and power consumption

## 3.2 Comparison with other structured-pruning methods

In this section, we report our key findings from applying ReplaceMe across various model architectures and benchmarks. To ensure the statistical stability of our results, all experiments are executed multiple times. As demonstrated in Fig. 2, we conduct a comparative analysis between ReplaceMe and UIDL [13], evaluating key metrics including time-to-get a comparable accuracy, environmental impact, and final model performance. Notably, for UIDL's healing process, we restricted fine-tuning to Low-Rank Adaptation (LoRA) applied exclusively to the MLP layers, whereas alternative approaches incur significantly higher computational costs. Our proposed method exhibits substantially reduced computational demands and achieves a markedly faster recovery compared to other methods. In Table 1, we compare ReplaceMe with other state-of-the-art structured depth pruning approaches. It should be noted that all competing methods rely on healing mechanisms and require extensive retraining, whereas our method remains completely training-free (no healing is applied). Despite this, as shown in Table 1, our method consistently outperforms all baselines on average and achieves 92.5% of the performance of the uncompressed Llama 2 7B model at a 25% compression ratio.

In Table 2, we further compare our method against state-of-the-art structured pruning approaches on the more recent Llama 3 8B Instruct model, under the setting where no healing is applied. We note that SVD-LLM [53] employs a low-rank approximation of the weights, while LLM-Pruner [29] combines both width and depth pruning. Despite these differences, as shown in Table 2, our method again outperforms these baselines. All results are reported at a 25% compression ratio. The Multi_LT results correspond to the application of the method described in Section 2.4. While the identification of multiple groups of blocks typically yields consecutive blocks in most cases, we additionally evaluate the scenario where non-consecutive blocks are enforced. This configuration demonstrates an improvement in perplexity but leads to a performance degradation across benchmark tasks.

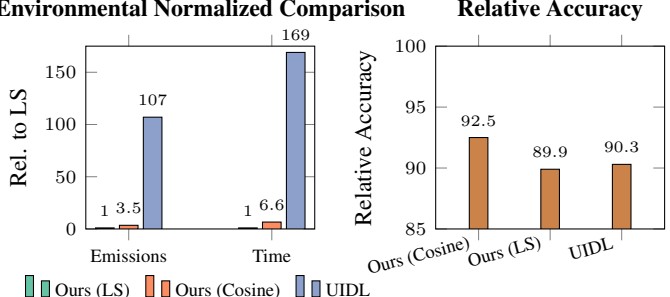

Figure 2: Comparison of the proposed LLM compression method with state-of-the-art UIDL. Subplots illustrate (a) compression time and environmental impact ($CO_2$ emissions), and (b) performance accuracy relative to the uncompressed baseline. Our approach attains the shortest compression time and reduced emissions, while achieving the highest accuracy, demonstrating superior efficiency, sustainability, and effectiveness over existing methods.

| Method | Train-Free | C3 | CMNLI | CHID (test) | WSC | Hella Swag | PIQA | Race-M | Race-H | MMLU | CMMLU | AVG | RP |
|---|---|---|---|---|---|---|---|---|---|---|---|---|---|
| Llama 2 7B (baseline) | | 43.8 | 33.0 | 41.6 | 37.5 | 71.3 | 78.1 | 33.1 | 35.5 | 46.8 | 31.8 | 45.3 | 100.0% |
| LLM-Streamline* | ✗ | **43.3** | 33.0 | 24.1 | 36.5 | **61.1** | **71.5** | 34.8 | 37.0 | 45.5 | 29.4 | 41.6 | 92.0% |
| LLMPruner* | ✗ | 29.7 | 33.4 | 28.4 | 40.4 | 54.6 | 72.0 | 22.9 | 22.0 | 25.3 | 25.0 | 35.4 | 78.2% |
| SliceGPT* | ✗ | 31.5 | 31.6 | 18.5 | 43.3 | 47.5 | 68.3 | 27.0 | 29.4 | 28.8 | 24.8 | 35.1 | 77.5% |
| LaCo* | ✗ | 39.7 | **34.4** | **36.1** | 40.4 | 55.7 | 69.8 | 23.6 | 22.6 | 26.5 | 25.2 | 37.4 | 82.7% |
| UIDL* | ✗ | 40.2 | **34.4** | 21.5 | 40.4 | 59.7 | 69.0 | 35.2 | 34.7 | 44.6 | 28.9 | 40.9 | 90.3% |
| Ours (Cosine) | ✓ | 42.5 | 33.0 | 25.2 | 38.5 | 59.4 | 71.1 | 35.4 | **36.7** | **46.4** | 30.4 | **41.9** | **92.5%** |
| Ours (LS) | ✓ | 39.4 | 33.0 | 18.9 | 38.5 | 58.5 | 70.5 | **37.1** | 36.5 | 45.2 | 29.2 | 40.7 | 89.9% |

Table 1: Comparing other pruning methods after healing and our training free approach ReplaceMe, * indicates that the numbers are taken from streamline paper [3]. After compressing Llama 2 7B with 25% compression ratio. ▮ signifies that the model was trained following pruning, whereas ▮ indicates that the model is training-free. We also report the relative performance to the original, unpruned model (RP).

In Figure 3, we also compare our training-free approach, ReplaceMe, against UIDL [13] across various models and different amounts of layer pruning. Our method consistently outperforms UIDL in both benchmark scores and perplexity evaluations, while also exhibiting greater stability.

Finally, we note that at high compression ratios, applying a healing process becomes necessary, as linear transformations alone are insufficient to fully recover performance. Details are provided in Table 3. Although ReplaceMe continues to outperform UIDL under these conditions, a healing phase is required at extreme compression levels to maintain model effectiveness.

## 3.3 Analysis

Up to this point, we have outlined our primary goal: replacing a series of transformer blocks with a simpler, estimated linear transformation using calibration data. The nature of this calibration data is critical, as we demonstrate in Section 3.3.1, where the type of text (instructional vs. plain) and the amount of data significantly influence our results. Furthermore, in Section 3.3.2, we analyze the impact of regularization on results, revealing a trade-off between performance metrics such as perplexity and accuracy. In the supplementary material we further explore structured linear transformations (e.g., diagonal or orthonormal matrices) A.3.

### 3.3.1 Ablation on calibration data

Our pruning method eliminates the need for additional training by leveraging small calibration datasets in place of conventional training data. These calibration datasets serve two core purposes: 1) assessing block importance to identify transformer blocks for removal (Section 2.1), and 2) capturing hidden states before and after the pruned blocks to solve the optimization problem in Eq. 5, which leads to the estimated linear transformation. The quality and characteristics of these calibration subsets are critical to the accuracy of the estimation. To understand the influence of calibration data, we conducted ablation studies exploring the impact of sample size and dataset type—including plain text, instruction-tuned data, and self-generated content. Our primary experiments utilized datasets such as Arcee [5], FineWeb [36], and SlimOrca [26], consistent with prior work like UIDL [13].

| Method | Linear transform | Lambada OpenAI ppl ↓ | Avg-acc↑ | RP↑ |
|---|---|---|---|---|
| Llama 3 8B Instruct [8] | | 3.11 | 0.7 | 100% |
| SVD-LLM [53] | None | 29.90 | 0.59 | 85.3% |
| LLMPruner [29] | None | 12.31 | 0.60 | 85.3% |
| UIDL [13] | Identity | 2216.96 | 0.58 | 82.5% |
| ReplaceMe(ours) | Linear (LS) | 20.23 | 0.63 | 89.9% |
| ReplaceMe(ours) | Linear (Cosine) | **15.88** | **0.63** | **90.9%** |
| ReplaceMe(ours) | Multi_LT_NC (Cosine) | **13.95** | **0.63** | 90.0% |

Table 2: Results of pruning Llama 3 8B instruct for 25% using different methods, without any healing or finetuning. Avg-acc is the average performance across the Race, Winogrande, PIQA, BoolQ, OpenBookQA, and SciQ benchmarks. Perplexity is measured on the Lambada OpenAI dataset. We also report the performance relative to the original, unpruned model (RP). Multi_LT_NC denotes the non-consecutive blocks case when applying the method described in Section 2.4.

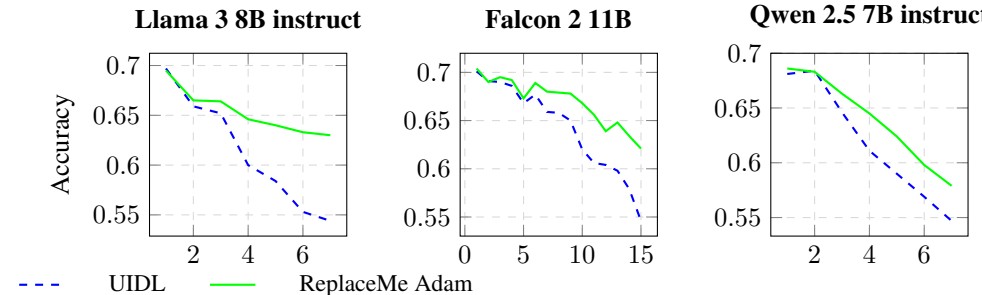

Figure 3: Comparison between our method and UIDL with different number of pruned layers and different models including, Llama 3 8B, Mistral 3 7B, and Qwen 2.5 7B. We show the average accuracy across the Race, Winogrande, PIQA, BoolQ, OpenBookQA, and SciQ benchmarks.

In particular, we investigated three key factors: (1) the effect of dataset type and its source, (2) the minimum amount of data required to produce stable and accurate estimates, and (3) the efficacy of masking as a lightweight data augmentation strategy when working with limited calibration samples. As shown in Table 4, we apply our method to prune LLaMA 3 8B [8] by 25%, using ReplaceMe across all ablation settings. We evaluate three distinct calibration datasets: FineWeb, a plain-text web corpus; SlimOrca, a curated instruction dataset generated with ChatGPT; and orca_generated, a synthetic dataset where responses are generated by the baseline model (targeted for pruning) using prompts from SlimOrca.

**Impact of Calibration Dataset Type** As shown in Table 4, calibration with instruction datasets leads to better performance on benchmark evaluations than using plain text, particularly for instruction-tuned models. While self-generated data yields improved perplexity scores, it tends to underperform on downstream benchmarks. This observation is consistent for both the optimization objectives that we utilize in Eq. 5, that is the $L_2$ and cosine distances.

We have also explored combining SlimOrca with other datasets, such as FineWeb and Aya [43]. The latter is a multilingual corpus covering 66 languages. Interestingly, these mixed datasets performed on par with SlimOrca, suggesting that high-quality instruction data have the biggest impact.

Furthermore, in the supplementary material (Section A.12), we present a comprehensive analysis of the impact of calibration dataset size, including the minimum number of samples required to achieve stable model performance. We also introduce a masking-based augmentation technique designed to

| Compression ratio | 12.5% | 25% | 37.5% | 50% |
|---|---|---|---|---|
| UIDL | 59.9 | 50.4 | 41 | 30.4 |
| Ours | **64.6** | **60.7** | **51** | **38.6** |

Table 3: Average accuracy for our method and UIDL at higher compression ratios is compared across the Race, Winogrande, PIQA, BoolQ, OpenBookQA, and SciQ benchmarks.

| Method | Objective | Calibration Data | Avg-acc ↑ | Perplexity ↓ | % ↑ |
|---|---|---|---|---|---|
| Baseline Model | - | - | 0.70 | 3.11 | 100.00 |
| ReplaceMe | LS | fineweb 8k | 0.56 | 26.74 | 80.47 |
| ReplaceMe | LS | slim_orca 8k | **0.62** | 21.21 | **89.59** |
| ReplaceMe | LS | orca_generated 8k | 0.61 | **13.58** | 87.40 |
| ReplaceMe | Cosine | fineweb 8k | 0.58 | 25.07 | 83.16 |
| ReplaceMe | Cosine | slim_orca 8k | **0.63** | 15.90 | **90.67** |
| ReplaceMe | Cosine | 4K SlimOrca + 4K Fineweb | 0.63 | 15.85 | 90.51 |
| ReplaceMe | Cosine | Mix of 66 languages | 0.63 | 15.72 | 90.64 |
| ReplaceMe | Cosine | orca_generated 8k | 0.61 | **13.24** | 87.33 |

Table 4: Pruning results for Llama 3 8B instruct using ReplaceMeby 25%. We estimate the linear transformation using different data, including plain text data such as Fineweb [36] and self generated data using Slim Orca instructions.

maintain robust performance even when only a limited subset of data samples is available, ensuring computational efficiency under resource-constrained conditions. In addition, Section A.17 presents results with task-specific calibration datasets, showing that using data from the same task as the evaluation target (e.g., SciQ) further improves accuracy compared to general-purpose calibration data such as SlimOrca.

### 3.3.2 Regularization effect

We have also investigated how regularization impacts the linear transform estimation. We have applied ridge regularization for the $L_2$ objective and observed that for $0 < \alpha < 10^3$ in Eq. 9, we notice a slight improvement in perplexity, while the average accuracy on benchmarks remains the same. Conversely, increasing $\alpha$ further to $10^4$ tends to enhance benchmark accuracy, though at the cost of higher perplexity. Thus, one can consider $\alpha$ as a tradeoff parameter between perplexity and accuracy of the pruned model. Regarding the cosine distance objective, $L_1$ regularization with $\alpha = 10^{-4}$ gives a higher boost to accuracy at the cost of perplexity performance. Similar results we obtain for $L_2$ regularization but compared to $L_1$ regularization, the performance gain on benchmarks is smaller.

| Model | Method | $\alpha$ | Avg-acc | Perplexity | RP |
|---|---|---|---|---|---|
| Llama3 8B | – | – | 0.697 | 3.1 | 100.0 |
| ReplaceMe | LS | 0 | 0.624 | 21.2 | 89.6 |
| ReplaceMe | LS + L2 reg | 0.1 | 0.625 | 21.2 | 89.6 |
| ReplaceMe | LS + L2 reg | 0.5 | 0.625 | 21.2 | 89.6 |
| ReplaceMe | LS + L2 reg | 10 | 0.624 | 21.2 | 89.5 |
| ReplaceMe | LS + L2 reg | 100 | 0.624 | 21.1 | 89.6 |
| ReplaceMe | LS + L2 reg | 1000 | 0.624 | **20.8** | 89.5 |
| ReplaceMe | LS + L2 reg | 10000 | **0.626** | 22.9 | **89.8** |
| ReplaceMe | Cosine | 0 | 0.634 | **15.9** | 90.9 |
| ReplaceMe | Cosine + L2 reg | 0.01 | 0.635 | 20.7 | 91.1 |
| ReplaceMe | Cosine + L1 reg | $1 \times 10^{-4}$ | **0.638** | 22.1 | **91.6** |

Table 5: The affect of regularization on LS and cosine methods in terms of accuracy and perplexity

### 3.4 Vision Transformers pruning

So far, we have focused on the application of ReplaceMe exclusively to the decoder transformer architecture, specifically within LLMs. This raises an important question: how well does this method generalize to other tasks beyond text generation, particularly when the transformer acts as an encoder. To answer this, we have applied ReplaceMe on the CLIP model for compression ratios of 13% and 25%. We utilized 8,000 samples from the MIMIC dataset and using the same evaluation procedure as in [37], we considered well-known benchmarks, namely MS-COCO [27], Cifar-10 [21], EuroSAT [15], VTAB [58], and Pascal VOC-2007 [9]. For comparison purposes with a state-of-the-art method we also report results when UIDL [13] is applied on the same model.

| Model | Compres. ratio | MS-COCO Captions (retrieval) | | Cifar10 (zero-shot) | | VOC2007 Multilabel (zero-shot) | VTAB/EuroSAT | |
|---|---|---|---|---|---|---|---|---|
| | | text recall@5 | vision recall@5 | acc1 | acc5 | mean_avg_p | acc1 | acc5 |
| CLIP-L/14 [37] | - | 0.794 | 0.611 | 0.956 | 0.996 | 0.790 | 0.625 | 0.960 |
| UIDL | 13% | 0.745 | 0.609 | 0.927 | 0.996 | 0.781 | 0.490 | 0.931 |
| ReplaceMe (LS) | 13% | **0.767** | **0.620** | **0.939** | 0.996 | **0.800** | **0.552** | **0.941** |
| UIDL | 25% | 0.515 | 0.418 | 0.693 | 0.971 | 0.597 | 0.381 | 0.814 |
| ReplaceMe (LS) | 25% | **0.556** | **0.471** | **0.780** | 0.971 | **0.688** | **0.395** | **0.823** |

Table 6: Pruning CLIP vision encoder using ReplaceMe. The model performance after compressing by 13% is almost as good as the original one, while in both scenarios our method outperforms UIDL.

As shown in Table 6, ReplaceMe retains a strong performance on CLIP-ViT [37] even at a 13% compression ratio, closely matching the original model's accuracy and without requiring any additional training. While performance declines at higher compression ratios, this degradation is expected and consistent across benchmarks. Despite this, ReplaceMe consistently outperforms the training-free state-of-the-art method UIDL [13]. Finally, we note that the performance can be further improved by utilizing a lightweight post-pruning "healing" procedure.

## 4 Related Work

Model pruning [23, 14] has been at the frontier of deep-learning research since the early developments of this field. It has found practical applications not only in model size reduction but also in enhancing the interpretability of the models under study. Earlier studies on transformer-based language models, such as BERT, demonstrated that these models are highly compressible [42, 40]. The same observation holds for pruning LLMs.

A significant number of studies focuses on unstructured pruning, where individual weights within matrices throughout the model are zeroed out, resulting in sparse connections. SparseGPT [11] tackles the challenge of layer-wise reconstruction in pruning by leveraging approximations of the inverse Hessian matrix. Wanda [47] improves the SparseGPT idea of reducing computations via simplification of the Hessian approximation. The LLM Surgeon [52] uses Kronecker-factored curvature approximations to perform pruning of LLMs. Despite maintaining high model quality post-pruning, computational savings from unstructured pruning requires specialized hardware support for sparse computations, limiting its wide applicability.

In contrast, structured pruning involves the complete elimination of certain structures inside the network. In this context, removing entire attention heads or MLP channels is referred to as width pruning. LLM-Pruner [29] suggested to calculate an importance metric based on the difference in the loss when this is computed with and without a pruned group of weights, respectively. FLAP [1] proposed a training free approach that is based on a fluctuation pruning metric and an adaptive compression ratio.

Another typical strategy within structured pruning is depth pruning. Methods in this category remove entire transformer layers of the network. In Shortened llama [18], the authors suggested identifying the significance of each decoder layer using perplexity analysis and a Taylor metric. This metric is based on a similar idea with the LLM-Pruner importance metric, that is, it measures the difference of the model loss when computed with and without a pruned layer. After pruning, the authors [18] further propose healing via LoRA fine-tuning, continual pre-training, or their combination.

The authors of ShortGPT [30] introduced the Block Influence (BI) metric to quantify the contribution of each network layer. This metric corresponds to the cosine distance between the hidden states before and after the layer. After pruning, they optionally recommend retraining the model to recover the model's performance. In contrast, UIDL [13] suggested computing the importance of a fixed-length sequence of layers instead of computing this metric for each layer individually. They calculate the cosine distance between the input and the output of the sequence and then if the distance is below a pre-defined threshold they remove the entire sequence of layers completely. Post-removal, healing with LoRA on the MLP is applied. In the recent LLM-Streamline paper [3], the authors propose to replace a fixed-length sequence of layers with a lightweight network, which can be either a transformer layer or a Feed-Forward Network (FFN). This network is then trained using the MSE loss and the LLM loss with LoRA. Prior work [38] has demonstrated that certain transformer blocks exhibit

linear characteristics, with empirical analyses revealing near-perfect linear relationships between embedding transformations across layers. They suggest to substitute highly linear transformer blocks with linear layers, which facilitates efficient model compression through direct knowledge distillation. Furthermore, the study proposes incorporating a cosine-similarity-based regularization mechanism during pretraining to mitigate excessive linearity. Recently, the Minitron LLM family [33] and its pruned variants were introduced, demonstrating an effective balance between depth and width pruning to mitigate performance degradation. The approach estimates the relative importance of depth and width dimensions to achieve an optimal trade-off that minimizes performance loss. However, a notable drawback is its reliance on a substantial amount of data—approximately 100B tokens.

## 5  Limitations

While the effectiveness of our proposed method is mathematically justified and experimentally validated, it is important to note that due to the training-free nature of ReplaceMe, such effectiveness is primarily evident within certain compression ratio ranges. As demonstrated in Figure 3, ReplaceMe performs well without fine-tuning for what we refer to as "reasonable" compression ratios, which are dependent on both the size and architecture of the original model. For higher compression ratios, some degree of retraining or parameter adjustment becomes necessary. However, an advantage of our approach is that such finetuning can be limited to only the linear transformation, rather than requiring full-model fine-tuning. This selective healing preserves the computational efficiency of our method, as further detailed in the supplementary material, and contributes to its overall efficiency compared to alternative approaches.

## 6  Conclusion

In this work, we introduced a novel method named ReplaceMe, which is a training-free depth pruning method. Our proposed strategy involves substituting certain transformer blocks with a linear transform, which is estimated using calibration data. ReplaceMe requires no retraining or fine-tuning, yet it consistently outperforms existing pruning techniques in training-free settings and remains competitive even when compared to approaches that rely on post-pruning "healing" stages. We have conducted extensive experiments and outlined methodologies to accurately estimate linear transformations under different optimality criteria using both analytical and numerical techniques. The proposed method has been evaluated across a range of transformer architectures — including large language models and vision transformers — demonstrating its robustness, adaptability, and effectiveness. These results represent a significant step toward a truly training-free pruning strategy.

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

# A  Appendix

In this section, we revisit the central aspects of our research, starting with the closed-form solution presented in A.2. This provides a foundation for examining structured linear transformations (LTs) as discussed in A.3, leading to key experimental findings.

We evaluate multiple numerical solvers to minimize the cosine objective (A.6) and investigate the behavior of structured LTs. We then assess the applicability of our method on diverse model architectures, demonstrating the effect of the applied "healing" process on the proposed ReplaceMe. In addition, we include a computational comparison between ReplaceMe and competing approaches, as those introduced in our initial study.

Further, we dissect the role of integrating a linear layer as a standalone block between full layer activations. This is in contrast to our approach of merging LTs with MLP activation in the primary architecture. The obtained results reveal that both methods yield remarkably similar results.

Finally, to validate the robustness and generalization ability of our method, we evaluate several models on a wide range of benchmarks. These experiments demonstrate the stability and adaptivity of our approach in varying conditions.

## A.1  Terminology and Definitions

- Relative Performance (RP): A normalized metric quantifying performance relative to the baseline model, computed as the ratio of benchmark results between the evaluated model and the baseline.

- Compression Ratio: The proportional reduction in model parameters expressed as a percentage, calculated as:

$$\left(1 - \frac{N_{pruned}}{N_{original}}\right) \times 100 \tag{10}$$

  where $N_{pruned}$ and $N_{original}$ represent the number of parameters in the compressed and original models, respectively.

- ReplaceMe: Our proposed method for model compression, which replaces complete transformer blocks with optimized linear transformations.

- Average Accuracy (Avg-acc): The arithmetic mean of model accuracy scores across multiple benchmark datasets.

- Perplexity: A metric for evaluating language model performance, defined as the exponential of the cross-entropy loss on the Lambada-openai benchmark.

## A.2  Closed-form solution for $L_2$ Distance

The optimization problem for the linear transform matrix estimation can be expressed as follows:

$$\mathbf{T}^* = \arg\min_{\mathbf{T}} \|\left(\mathbf{M}_i \cdot \mathbf{T} + \mathbf{Y}_i\right) - \mathbf{L}_{i+n}\|_2^2.$$

The objective is an Euclidean norm and thus we can expand it to:

$$\|\mathbf{R}\|_2^2 = \mathrm{tr}\left(\left(\mathbf{M}_i \cdot \mathbf{T} + \mathbf{Y}_i - \mathbf{L}_{i+n}\right)^\top \left(\mathbf{M}_i \cdot \mathbf{T} + \mathbf{Y}_i - \mathbf{L}_{i+n}\right)\right) \tag{11}$$
$$= \mathrm{tr}\left(\mathbf{T}^\top (\mathbf{M}_i^\top \mathbf{M}_i)\mathbf{T} + 2(\mathbf{Y}_i - \mathbf{L}_{i+n})^\top \mathbf{M}_i \cdot \mathbf{T} + \mathbf{L}_{i+n}^\top \mathbf{L}_{i+n}\right)$$

To minimize the objective, we take the gradient with respect to $\mathbf{T}$ and set it to zero:

$$\nabla_{\mathbf{T}}\|\mathbf{R}\|_2^2 = 0 \tag{12}$$
$$2\mathbf{M}_i^\top \mathbf{M}_i \cdot \mathbf{T} + 2\mathbf{M}_i^\top (\mathbf{Y}_i - \mathbf{L}_{i+n}) = 0$$
$$\mathbf{M}_i^\top \mathbf{M}_i \cdot \mathbf{T} = \mathbf{M}_i^\top (\mathbf{L}_{i+n} - \mathbf{Y}_i)$$
$$\boxed{\mathbf{T}^* = \left(\mathbf{M}_i^\top \mathbf{M}_i\right)^{-1}\mathbf{M}_i^\top (\mathbf{L}_{i+n} - \mathbf{Y}_i)}$$

## A.3 Structured LT Matrix

To improve the interpretability of our approach, we further investigate additional constraints that can be imposed on the structure of the linear transformation $\mathbf{T}$. The results of all the different constrained transformations are presented in Sec A.4. Motivated by the Transformers-squared work [48], we also consider the case where we condition $\mathbf{T}$ to be a diagonal matrix. This constraint is meaningful under the assumption that an adequate mapping of the activations is possible through only scaling of the hidden states. In this case, the optimization problem is of the form:

$$\mathbf{T}^* = \arg\min_{\mathbf{T} \in \mathcal{D}^{d \times d}} \|(\mathbf{M}_i \cdot \mathbf{T} + \mathbf{Y}_i)) - \mathbf{L}_{i+n}\|_2^2,$$

where $\mathcal{D}^{n \times n}$ denotes the space of diagonal matrices of dimensions $d \times d$. The corresponding closed-form solution is given by:

$$\mathbf{T}^* = \left((\mathbf{M}_i^\top \mathbf{M}_i) \circ \mathbf{I}\right)^{-1} \left((\mathbf{M}_i^\top (\mathbf{L}_{i+n} - \mathbf{Y}_i)) \circ \mathbf{I}\right), \tag{13}$$

where $\circ$ denotes the Hadamard product.

Another constraint that can be imposed on $\mathbf{T}$ is the requirement for it to represent an orthogonal transformation [51]:

$$\mathbf{T}^* = \arg\min_{\mathbf{T}} \|(\mathbf{M}_i \cdot \mathbf{T} + \mathbf{Y_i}) - \mathbf{L}_{i+n}\|_2^2 \quad \text{s.t.} \quad \mathbf{T}^\top \mathbf{T} = \mathbf{I}.$$

This problem admits an analytical solution via singular value decomposition. Specifically, let $\text{SVD}(\mathbf{M}_i^\top (\mathbf{L}_{i+n} - \mathbf{Y_i})) = \mathbf{U} \cdot \mathbf{\Sigma} \cdot \mathbf{V}^\top$, then the optimal orthonormal matrix is given by:

$$\mathbf{T}^* = \mathbf{U} \cdot \mathbf{V}^\top.$$

## A.4 Results of Structured Linear Transformations

Our evaluation protocol, summarized in Table 7, explores constrained forms of the linear transformation matrix $\mathbf{T} \in \mathbb{R}^{d \times d}$. Inspired by the architectural design of Transformer-squared [48], we first examine the case where $\mathbf{T}$ is restricted to a diagonal matrix, i.e., $\mathbf{T} = \text{diag}(t_1, \ldots, t_d)$. Although this parameterization successfully restored model functionality, the resulting perplexity $\mathcal{P}$ remained suboptimal, with $\mathcal{P}_{\text{diag}} > \mathcal{P}_{\text{generic}}$, where $\mathcal{P}_{\text{generic}}$ represents the baseline achieved using an unconstrained full matrix.

| Model | LT Structure | Multi-linear Transform | Avg-acc ↑ | Perplexity ↓ | % ↑ |
|-------|--------------|------------------------|-----------|--------------|-----|
| Llama3 8B | - | - | 0.70 | 3.11 | 100.00 |
| UIDL | - | ✗ | 0.58 | 2216.96 | 82.5 |
| ReplaceMe | Generic | ✗ | **0.62** | **21.21** | **89.59** |
| ReplaceMe | Orthonormal | ✗ | 0.60 | 700.57 | 85.67 |
| ReplaceMe | Diagonal | ✗ | **0.62** | 89.09 | 88.42 |
| ReplaceMe | Generic | non-consecutive | 0.62 | **16.07** | 89.60 |

Table 7: Ablation study on (un)constrained LT matrix structure (Generic, Diagonal, Rotational) and multi-linear transforms. The base model is set to Llama 3 8B with 25% pruning ratio.

Subsequent experiments with orthogonal transformations $\mathbf{T}$ (where $\mathbf{T}^\top \mathbf{T} = \mathbf{I}$) demonstrated limited efficacy, yielding only marginal performance recovery. This indicates that actually **scaling is much more important to recover model performance than rotations or reflections**.

More promising results emerged from employing multiple linear transformations $\{\mathbf{T}_i\}_{i=1}^k$. We examined the following approach: **Non-consecutive transformations**: For disjoint parameter subsets, we learned independent transformations $\{\mathbf{T}_i^{\text{disjoint}}\}_{i=1}^k$. The experimental results indicate that applying multiple transformations yields consistent improvements in both perplexity ($\mathcal{P}$) and accuracy ($\mathcal{A}$) across both analytical and numerical solutions. For the analytical approach, these gains are achieved in a single computation step , whereas the numerical solution requires iterative optimization, leading to significantly higher computational costs.

### A.5    Statistical Significance

To ensure reproducibility and stability of our method we ran our method multiple times for Llama-3-8B-Instruct with 8 pruned layers. While the analytical solution always leads to the same results, numerical methods might have different results. Howewer, we were able to get almost the same results: (1) the mean average accuracy over all benchmarks equals to 60.71 and the standartd deviation (std) is 0.08; (2) the mean lambda-openai perplexity equals to 15.90 and the standartd deviation (std) is 0.05.

### A.6    Comparative Analysis on the Performance of Numerical Solvers

This section investigates the effects of different optimization solvers when minimizing the cosine distance objective function. The previously reported results were obtained using the Adam optimizer; here, we perform a systematic comparison using alternative solvers, including: 1) Non-Linear Conjugate Gradient (NLCG), Newton-CG, and Limited-memory BFGS (L-BFGS). Additionally, we compare the results obtained when minimizing the Mean Squared Error (MSE) either via numerical optimization or using the Least Squares (LS) closed-form solution. These comparisons provide insights about the solver efficiency and the solution accuracy for different objective functions. As

| Solver | Avg-acc | RP | Percentage |
|---|---|---|---|
| Baseline[19] | **0.697** | **3.106** | **100** |
| Adam[19] | **0.634** | **15.875** | **90.940** |
| NLCG[34] | 0.630 | 15.960 | 90.412 |
| L-BFGS[28] | 0.634 | 16.200 | 90.921 |
| Trust-NCG[45] | 0.620 | 47.000 | 88.945 |
| LS | 0.624 | 21.206 | 89.588 |
| MSE + Adam | 0.624 | 20.633 | 89.536 |

Table 8: Comparison between different solvers to estimate the linear transform for Llama 3 8B after 25% compression with cosine distance objective, and a sanity check between MSE with Adam solver as a numerical solution and LS analytical solution

demonstrated in Table 8, nearly all solvers converge to a similar point, with results on both benchmarks and perplexity metrics being very close. An exception is the Trust-NCG solver, which seems to got trapped in a local minimum. Furthermore, there is a clear indication that miminizing the MSE, either numerically or analytically, yields nearly identical outcomes. Nevertheless, the analytical solution is much faster and has much less hardware requirements.

### A.7    Generalization Across Model Scales

While the primary results presented in this work focus on a subset of model architectures, these findings may not fully capture the scaling behavior across a broader parameter spectrum, where model sizes can vary from $(1)$ billion to $(70)$ billion parameters or beyond. To rigorously investigate the scaling properties of our approach, we conduct extensive experiments across the following benchmark suite: (Winogrande [41], BoolQ [4], OpenBookQA [31], SciQ [55], Race [22] and PIQA [2]). Our scaling analysis involves the Llama 3 architecture family, from the compact Llama 3.2 ($1B$ parameters) to the largest available variant, Llama 3 ($70B$ parameters). As evidenced in Table 9, we

| Model | Llama-3.2-1B-Instruct | | Llama-3.1-8B-Instruct | | Llama-3-70B-Instruct | | |
|---|---|---|---|---|---|---|---|
| | Baseline | ReplaceMe | Baseline | ReplaceMe | Baseline | ReplaceMe | ReplaceMe |
| Avg-Acc | 0.6098 | 0.5348 | 0.7118 | 0.6537 | 0.728 | 0.7036 | 0.6596 |
| RP | 1.0 | 0.8771 | 1.0 | 0.9184 | 1.0 | 0.9664 | 0.9060 |
| Compression Ratio | 0% | 25% | 0% | 25% | 0% | 25% | 37.5% |

Table 9: Results of utilizing ReplaceMe on various model sizes, characterized by the number of parameters, for the Llama 3 Families. For Llama-3-70B Instruct we show ReplaceMe results with 25% and 37.5% compression ratios.

observe a strong correlation between model size and achievable compression ratio $\eta$. Specifically, for

large-scale models ($70B$ parameters), we achieve optimal compression ratios of $\eta_{\max} = 37.5\%$ while maintaining performance retention $\mathcal{RP} \geq 90\%$ of the original model's capability.

## A.8 Computational Efficiency Analysis

This section presents a quantitative comparison between our training-free method and the baseline UIDL framework, with a focus on computational overhead, energy consumption, and associated $CO_2$ emissions. As demonstrated in the Table 1, our approach achieves competitive performance despite being training-free, whereas UIDL requires fine-tuning on a limited dataset. To ensure an equitable comparison, we adopt this configuration to rigorously evaluate the computational advantages of our method. Furthermore, we conduct a comparative analysis between our proposed method with partial healing and the complete model healing approach. As demonstrated in Section A.14, our methodology requires only the healing of the LT component while maintaining competitive performance relative to the full-model healing. For the UIDL baseline, we adhere to the original implementation of the authors, which employs LoRA-based healing exclusively on the MLP layers. As demonstrated in Fig.

Figure 4: Comparison between ReplaceMe and UIDL in terms of computation and environmental impact.

4, a comparative analysis of our two proposed methodologies is presented. The LS approach offers significant advantages in terms of cost-efficiency, speed, and the reduced demand for computational resources and memory. However, this method incurs a slight decline in performance metrics. Thus, the choice between these methods should be informed by the specific requirements and available hardware resources of the user.

## A.9 Mergable LT vs. LT as an Independent Block

| LT | Objective | Avg-accuracy | Perplexity | RP |
|---|---|---|---|---|
| Fusable into MLP | Cosine | 0.634 | 15.875 | 0.905 |
| Separate LT block | Cosine | 0.640 | 13.481 | 0.913 |

Table 10: Compressing Llama 3 8B by 25% using a cosine distance objective, without applying healing. This involves applying LT on the MLP output or on the full output activation of the transformer block.

In our work, we propose integrating the linear transformation (LT) matrix so that it can be merged with the down-projection component of the MLP layer, preserving architectural compatibility with standard LLM designs. However, this approach raises a critical question regarding the comparative efficacy of injecting the LT as a mergable layer versus incorporating it as an independent, non-mergable block within the transformer architecture. In Table 10, we present the outcomes of integrating the linear transform (LT) directly into the Multi-Layer Perceptron's (MLP) down-projection through matrix fusion. This approach ensures parameter efficiency in contrast to employing a standalone LT Block, where the LT functions as an independent layer between transformer blocks, thereby introducing additional computational overhead. The results demonstrate a marginal improvement; however, the extent of this improvement is contingent upon user-specific requirements.

## A.10   Block Selection Analysis

In this section, we conduct a systematic ablation study to investigate the relationship between inter-layer activation distance metrics and model performance under sequential layer removal. We employ a sliding window approach, iteratively pruning contiguous blocks of eight layers starting from the initial layer index and incrementally shifting the window by a single layer position. For each configuration, we quantify activation dissimilarity using both cosine distance and L2 norm , computed between intermediate activations of the original and pruned models. Subsequently, for our method we estimate the linear transformation (LT) and evaluate the pruned architectures on a standardized subset of benchmark tasks. As demonstrated in Fig. 5, we observe a strong inverse

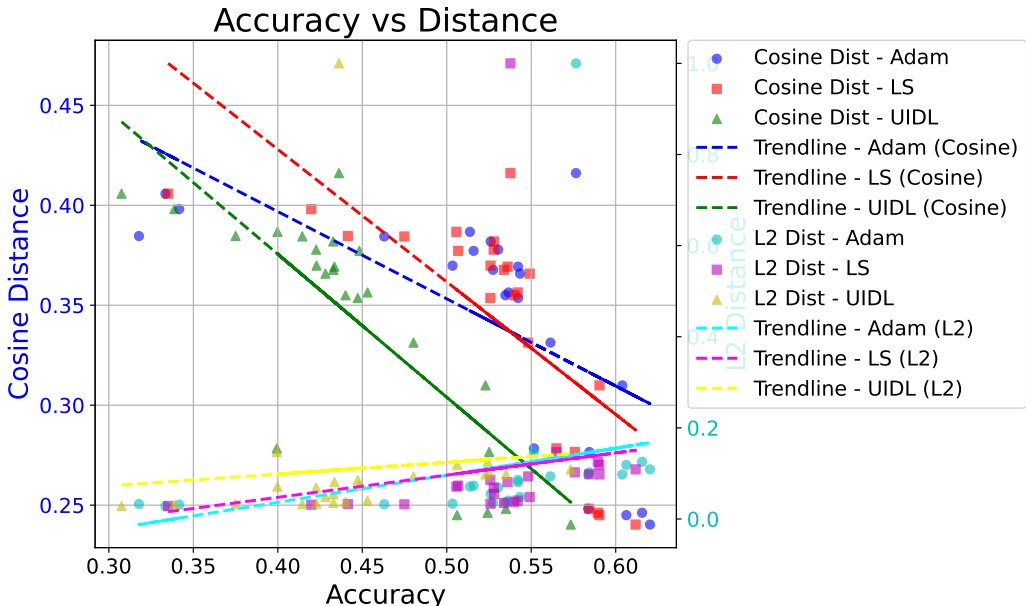

Figure 5: Comparative analysis of distance metrics and predictive accuracy across layer pruning configurations. Trendlines illustrate the inverse correlation between cosine distance and accuracy, contrasted with the positive correlation between L2 and accuracy degradation . Results are shown for ReplaceMe with LT estimation via Cosine/LS metrics and UIDL baselines.

correlation between cosine distance reduction and accuracy improvement across all methodologies. Specifically, ReplaceMe with cosine-based LT estimation achieves peak performance at the lowest cosine distance values, Conversely, configurations exhibiting lower L2 correspond to significant accuracy degradation suggesting that L2 lacks power for optimal layer selection.

In Fig. 6, we present the detailed results of sequential layer removal and compare the cosine distance with other similarity measures: CKA [20] and CCA [32]. We observe that the highest accuracy corresponds to the lowest cosine distance, which confirms the validity of using cosine distance as a block selection criterion.

It is also important to note that the optimal pruning range depends on both the architecture and the number of layers removed. Previous works have also observed that layers form natural groups [49]. We conducted a similar experiment and plotted the cosine distance across all layers in Fig. 7.

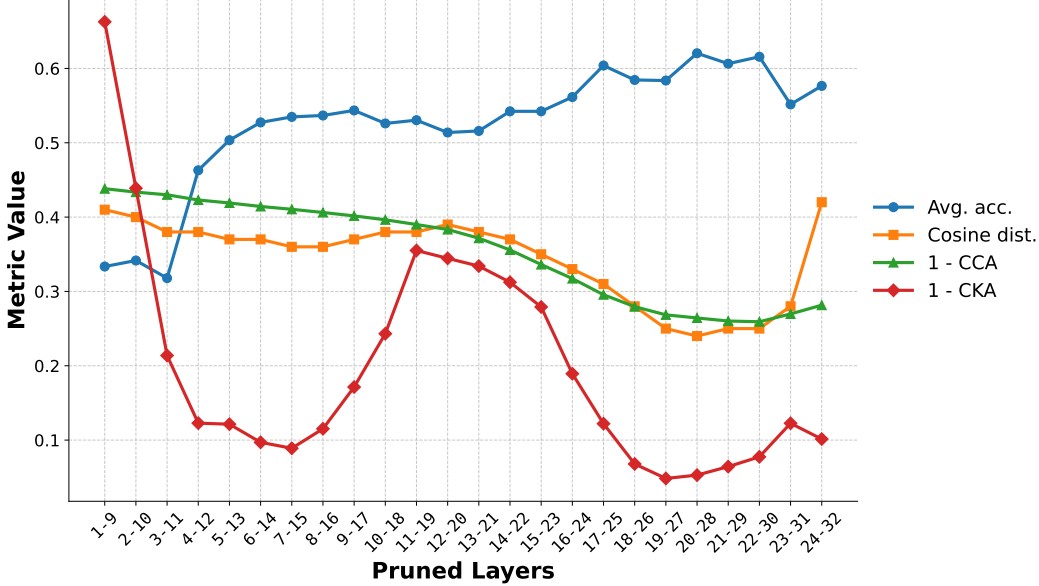

Figure 6: Comparison of cosine distance, CCA, and CKA across all 8-layer pruning configurations in Llama3 8B.

## A.11 Cosine Distance Approximation

In this section, we discuss further our proposed approximation of the loss, where we subtract the attention output from the activation after the cut, when using the cosine distance. First, we revisit the original Eq. (14)

$$\mathbf{T}^* = \arg\min_{\mathbf{T}} \cos\left(\mathbf{M}_i \cdot \mathbf{T} + \mathbf{Y}_i, \mathbf{L}_{i+n}\right), \tag{14}$$

where we observe that in order to estimate $\mathbf{T}$ numerically we need to store 3 activations $(\mathbf{M}_i, \mathbf{Y}_i, \mathbf{L}_{i+n})$ for each token. This is not effective and requires considerable memory and time to compute. To overcome this issue, we apply the cosine distance after subtracting the attention output from the full activation at the end-block of the cut as formulated in Eq.(15):

$$\mathbf{T}^* = \arg\min_{\mathbf{T}} \cos\left(\mathbf{M}_i \cdot \mathbf{T}, \mathbf{L}_{i+n} - \mathbf{Y}_i\right). \tag{15}$$

As evidenced by the experimental results presented in Table 11, the approximated cosine formulation

| Model | Method | Pruned Layers | Calibration Data | Training State | Perplexity | Avg-acc |
|---|---|---|---|---|---|---|
| Llama 3 8B instruct | approx. loss | 8 | slim_orca | no training | **15.88** | **0.634** |
| Llama 3 8B instruct | orig. loss | 8 | slim_orca | no training | 16.63 | 0.630 |
| Qwen 2.5 7B instruct | approx. loss | 7 | slim_orca | no training | **7.92** | **0.591** |
| Qwen 2.5 7B instruct | orig. loss | 7 | slim_orca | no training | 10.60 | 0.580 |
| Llama 3 8B instruct | Multi_A approx. loss | 8 | slim_orca | no training | 13.95 | 0.628 |
| Llama 3 8B instruct | Multi_A orig. loss | 8 | slim_orca | no training | **13.12** | **0.630** |

Table 11: Comparison between the original loss and the approximated version. The approximated version makes use of the cosine distance between the transformed MLP output and the residual of the activation of $(i + n)$-th block and the attention of the $i$-th block.

achieves comparable performance to the exact loss computation, with marginal improvements observed. This approximation yields memory efficiency gains, requiring only two stored activations per token $(\mathbf{M}_i, \mathbf{L}_{i+n} - \mathbf{Y}_i)$, thereby reducing memory usage by approximately 66% relative to the original implementation.

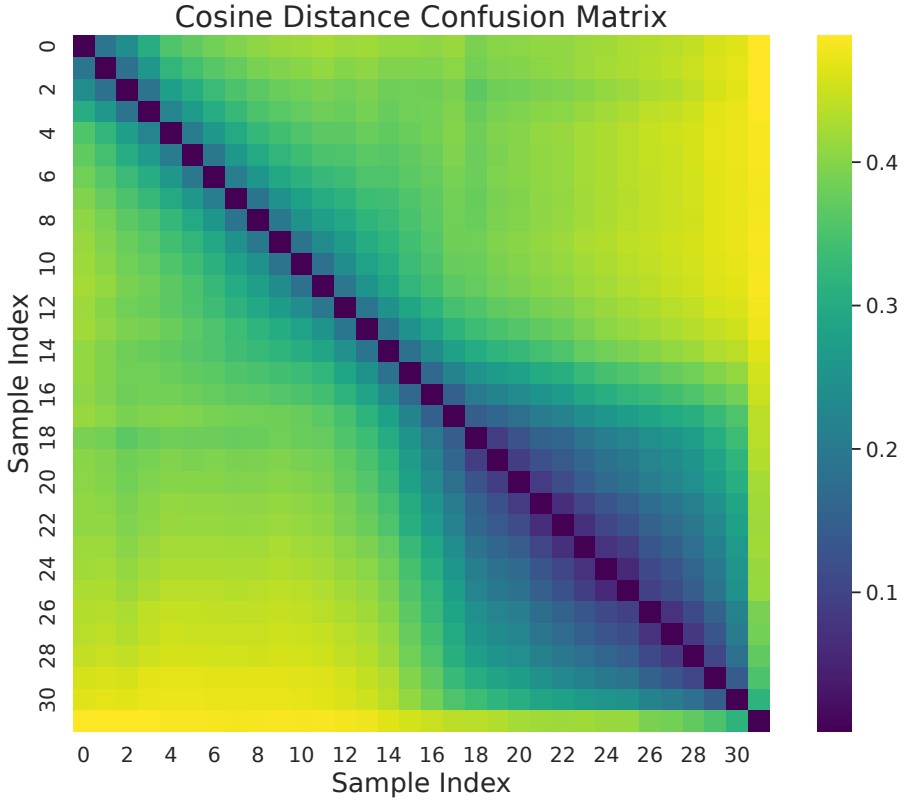

Figure 7: Average cosine distance between hidden states of layers for Llama 3 8B. Each cell $(i, j)$ shows the mean cosine distance between the output vectors of $layer_i$ and $layer_j$, computed across 256 samples of SlimOrca.

## A.12 Data ablation

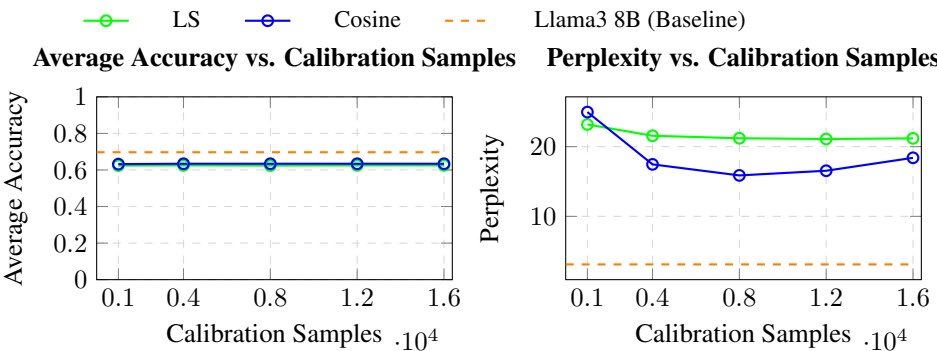

Figure 8: Pruning Llama 3 8B by 25% using different number of samples to estimate the linear transform

**Impact of Calibration Dataset Size** Figure 8 illustrates how the size of the calibration dataset affects the linear transform estimation, for both the $L_2$ and the cosine distance objectives. Although increasing the number of calibration samples does not significantly improve benchmark accuracy, it does substantially reduce perplexity, especially when the cosine distance objective is considered.

The linear transformation matrix has shape $d \times d$, requiring approximately $N = d^2$ tokens for reliable estimation. With a per-sample sequence length of $S$, this translates to at least $d^2/S$ samples. For example, in LLaMA 3 8B, where $d = 4096$ and $S = 1024$, about 16,000 samples are theoretically

needed. However, as seen in Fig 8, accuracy remains stable even with as few as 1,000 samples, suggesting robustness to sample size. That said, perplexity continues to improve with more data, indicating better model confidence and predictive quality.

**Impact of Masking for Data Augmentation** We have also investigated random token masking as a lightweight data augmentation technique for scenarios with limited calibration data (e.g., 1,000 samples). As shown in Table 12, masking improves the stability of the numerical optimization and leads to better convergence when estimating the linear transformation. This masking strategy proves especially beneficial in low-data regimes, where it reduces overfitting and enhances generalization. However, when more data are available, the impact of masking becomes negligible.

In summary, a calibration dataset with approximately $d^2$ tokens ensures stable and accurate estimation. When working with fewer tokens, random masking can mitigate overfitting and improve estimation quality. For instruction-tuned models, instruction-style calibration data consistently leads to better pruning outcomes. While self-generated data can reduce perplexity, it may degrade benchmark accuracy, highlighting a trade-off between confidence and task-specific performance.

| Model | Masking | Calibration Data | Avg-acc ↑ | Perplexity ↓ | % ↑ |
|---|---|---|---|---|---|
| Llama3 8B | - | - | 0.697 | 3.10 | 100.00 |
| ReplaceMe | ✗ | 1k | 0.632 | 24.96 | 90.62 |
| ReplaceMe | ✓ | 1k | 0.634 | **21.08** | **90.91** |
| ReplaceMe | ✗ | 8k | 0.633 | **15.88** | **90.76** |
| ReplaceMe | ✓ | 8k | 0.632 | 15.69 | 90.73 |

Table 12: Random token masking during the estimation of the linear transformation contributes to a more stable solution, particularly when working with small datasets.

## A.13 Extra Model Evaluation

To further validate the efficacy of our pruned models, we conducted an additional evaluation on a new subset of benchmarks derived from the Huggingface Leaderboard [10], utilizing a modified version of Eval-Harness [12]. This benchmark set encompasses Big Bench Hard (BBH) comprising 23 complex and diverse tasks [50]; High-school-level mathematical competition problems [17]; PhD-level domain expertise assessments across disciplines [39]; Algorithmically generated intricate problem sets [44]; A refined version of (MMLU) benchmark [54].

As shown in Fig. 9, the normalized results demonstrate that the Llama2 model preserves performance across nearly all benchmarks post-compression. Notably, Llama3 variant has a marginal performance degradation on the mathematical benchmark, potentially attributable to task-specific sensitivity to parameter reduction. Depending on the user's specific needs, an additional healing process can be considered to overcome this issue.

## A.14 Healing Experiments

Here, we present a comparative evaluation of our method in three experimental configurations: (1) the baseline implementation without any healing mechanism, (2) partial healing of only the learned linear transformation (LT) layer, and (3) complete model fine-tuning using a subset of the C4 dataset.

| Method | Training Params | Race | Winogrande | PiQA | BoolQ | OpenBookQA | SciQ | Perplexity | Avg. Acc |
|---|---|---|---|---|---|---|---|---|---|
| Llama 2 7B | None | 0.3952 | 0.7427 | 0.7900 | 0.7783 | 0.4400 | 0.9050 | 3.3973 | 0.6752 |
| ReplaceMe | None | 0.3656 | 0.6977 | 0.7051 | 0.7263 | 0.3460 | 0.8370 | 22.3586 | 0.61295 |
| ReplaceMe | Only LT | 0.3799 | **0.7182** | 0.7350 | 0.7728 | 0.3760 | **0.8840** | 5.3965 | 0.64432 |
| ReplaceMe | Full model | **0.3847** | 0.7056 | **0.7416** | **0.7835** | **0.3920** | 0.8720 | **4.9138** | 0.64657 |

Table 13: Performance comparison of ReplaceMe with/without healing and with different trainable parameters.

As demonstrated in Table 13, the model variants which involve healing yield performance improvements. Notably, the LT-only healing approach achieves comparable accuracy to the full model

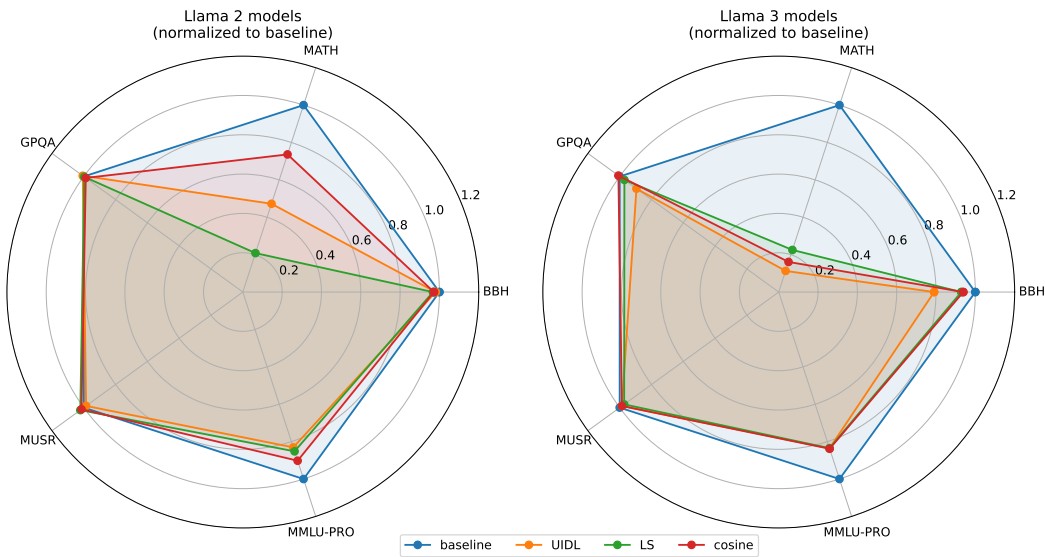

Figure 9: Comparison between our proposed methods and UIDL on a different set of benchmarks. Results are normalized relative to the baseline model performance.

fine-tuning while offering substantially reduced computational costs. This selective healing strategy demonstrates superior efficiency compared to alternative healing approaches documented in prior work.

### A.15 Computation environment

All experiments were conducted using an **NVIDIA A100-SXM4-40GB** GPU with an **AMD EPYC 7742 64-Core Processor**, running Ubuntu 22.04 and Python 3.10. The software environment was based on the official NVIDIA PyTorch container `nvcr.io/nvidia/pytorch:23.10-py3`.

For additional testing and validation, models were also tested on a **P100 GPU** using the Kaggle environment, which imposes stricter compute and memory constraints.

The computational setup and training configurations are summarized in two tables: Table 14 details the hyperparameters for ReplaceMe(cosine), and Table 15 compares the two healing experiments: training only LT versus full-model fine-tuning.

| Parameter | Value |
|---|---|
| Optimizer | Adam |
| Learning Rate | 0.0001 |
| Batch Size | 1024 |
| Epochs | 10 |
| Loss Function | Cosine Distance Loss |
| Weight Initialization | Identity Matrix |
| Bias | False |

Table 14: Hyperparameters and Configuration for ReplaceMe(Cosine-based Training)

### A.16 Multi-Linear transformations

To isolate the impact of number of LTs, we fix the total pruned layers at eight and vary how many of those are implemented as LT: 1, 2, 4, or 8, then benchmark both Llama-3-8B-Instruct and Mistral-7B-Instruct-v0.3 on the same benchmarks. We approximate using ReplaceMe(LS) method. Table 16 summarizes the results.

| Parameter | Only LT Training | Full Model Training |
|---|---|---|
| Optimizer | Adam | Adam |
| Context Length | 2048 | 2048 |
| Learning Rate | 3e-4 | 1e-5 |
| Number of GPUs | 1 | 4 |
| Batch Size (per device) | 1 | 1 |
| Max Steps | 80,000 | 20,000 |
| Gradient Accumulation Steps | 1 | 1 |
| Gradient Checkpointing | True | True |
| Unfrozen Weights | LT Only | Full Model |
| Trainable Parameters | 16.8M | 6.3B |

Table 15: Hyperparameters for Healing Experiments (`full_transform` vs. `full_model`)

| Method | Number of LTs | Race | Winogrande | PiQA | BoolQ | OpenBookQA | SciQ | Perplexity | Avg. Acc |
|---|---|---|---|---|---|---|---|---|---|
| Llama-3-8B-Instruct | | | | | | | | | |
| ReplaceMe | 1 | 0.3694 | 0.7167 | 0.6844 | 0.8061 | 0.3300 | 0.8400 | 21.2061 | 0.6244 |
| ReplaceMe | 2 | 0.3885 | 0.7261 | 0.6872 | 0.7798 | 0.3380 | 0.8580 | 18.9853 | **0.6296** |
| ReplaceMe | 4 | 0.3751 | 0.7277 | 0.6953 | 0.7661 | 0.3240 | 0.8590 | **16.0669** | 0.6245 |
| ReplaceMe | 8 | 0.3876 | 0.7017 | 0.6834 | 0.7165 | 0.3360 | 0.8300 | 37.9760 | 0.6092 |
| Mistral-7B-Instruct-v0.3 | | | | | | | | | |
| ReplaceMe | 1 | 0.4105 | 0.7530 | 0.6893 | 0.8560 | 0.3400 | 0.8780 | 9.8590 | 0.6545 |
| ReplaceMe | 2 | 0.4134 | 0.6156 | 0.7486 | 0.6985 | 0.3580 | 0.9020 | 7.1119 | 0.6227 |
| ReplaceMe | 4 | 0.4182 | 0.7119 | 0.7040 | 0.8287 | 0.3640 | 0.9140 | **5.9130** | **0.6568** |
| ReplaceMe | 8 | 0.4077 | 0.6551 | 0.7318 | 0.8174 | 0.3820 | 0.9120 | 5.9897 | 0.6510 |

Table 16: Performance comparison of ReplaceMe(LS) with the different number of LTs. We fix a total of 8 pruned layers and vary how many LTs we insert. Reported are task accuracies on the same set of benchmarks for both Llama-3-8B-Instruct and Mistral-7B-Instruct-v0.3.

For both models, using two or four LT modules tends to offer the best trade-off between task performance and language modeling quality. However, the differences across configurations are relatively small, and results are largely comparable. Fewer LT modules may under-adapt the pruned model, while more modules can introduce unnecessary complexity. The adaptation method appears robust to moderate variation in the number of LT modules.

## A.17  Task specific LT

We ran an additional experiment to see how the choice of calibration data affects results. In this test, we used the training part of the SciQ dataset as a calibration data and then measured performance on its test set. For comparison, we also applied our method using part of the general SlimOrca dataset but still evaluated it on SciQ. Using the Llama3 8B model, we found that calibration data from the same task leads to much better accuracy than calibration with general-purpose data. This shows that tailoring calibration data to the target task can significantly improve compressed model performance. See the results in Table 17.

| Model | Calibration Data | Compression | Sciq Accuracy |
|---|---|---|---|
| Llama3 8B instruct | - | - | 0.93 |
| UIDL | Sciq- Task specific | 25% | 0.687 |
| Ours (LSTSQ) | Sciq- Task specific | 25% | **0.89** |
| Ours (LSTSQ) | Orca General | 25% | 0.858 |

Table 17: Performance comparison of Llama3 8B models calibrated with task-specific (SciQ train split) versus diverse open-source (Orca subset) datasets on the SciQ test split

