# OpenReview forum: "ReplaceMe: Network Simplification via Depth Pruning and Transformer Block Linearization"
_NeurIPS.cc/2025/Conference — NeurIPS 2025 poster_

### Official Review · Reviewer_PYR3 · 2025-06-13

**Clarity:** 4
**Significance:** 3
**Originality:** 3
**Rating:** 5
**Confidence:** 3

**Summary:**

This paper proposes a pruning method that is both data- and computation-efficient. The proposed ReplaceMe substitutes selected Transformer blocks with linear layers, guided by an importance metric. This strategy aims to reduce model complexity while minimizing performance degradation, serving as an alternative to directly removing blocks. The authors present experimental results indicating that ReplaceMe can match or even outperform structured pruning methods that require retraining.

**Questions:**

I am curious about the size of the added linear transformations used for error compensation. It would be useful to clarify whether the number of additional parameters introduced by these layers has a significant impact on overall performance or model size. A discussion or ablation on this aspect could enhance the understanding of the method’s trade-offs.

**Ethical Concerns:**

["NO or VERY MINOR ethics concerns only"]

**Final Justification:**

I keep my rating since I did not pose too many questions towards this work, and the authors solved my concerns about specific experimental settings and provided additional experiments.

**Limitations:**

Yes.

**Paper Formatting Concerns:**

No paper formatting concerns.

**Quality:**

3

**Strengths And Weaknesses:**

# Strength #

* The paper addresses a practical and important topic in model compression. Structural pruning is generally more hardware-friendly but often more challenging in terms of preserving model performance. Thus, research that advances this area is both timely and valuable. Furthermore, the emphasis on pruning efficiency aligns well with real-world deployment constraints.

* The paper is clearly written and well-organized, making it easy to follow the proposed method and its motivation.

* The proposed method is both effective and efficient. The key idea of compensating for pruning-induced errors by inserting linear layers is compelling. These linear layers are data-efficient to train due to their convex nature and can be integrated into existing MLP layers without increasing inference cost, which enhances the practicality of the approach.

* The experiments span both NLP and CV tasks, demonstrating that the proposed method performs on par with or better than existing structured pruning techniques, including those that require additional tuning or retraining.

# Weakness #

I did not find a very strong weakness. Here are some minor suggestions.

* The first two subfigures in the introductory figure convey similar concepts and could potentially be merged to reduce redundancy and improve visual conciseness.

* It would be helpful to include an ablation study analyzing the distribution and quantity of skipped blocks. Specifically, showing how many layers are skipped at once and how this distribution affects performance would clarify how the redundancy lies.

---

> ### Author Rebuttal · Authors · 2025-07-30
>
> ### **Figure Modification**
> **Comment (1):** The first two subfigures in the introductory figure convey similar concepts and could potentially be merged to reduce redundancy and improve visual conciseness.
>
> **Response (1):** We thank the reviewer for his helpful comment. In the revised version of the paper we will combine and merge the two sub-figures.
>
> ### **Analysis Distribution and Quantity of Skipped Blocks**
> **Comment (2):** It would be helpful to include an ablation study analyzing the distribution and quantity of skipped blocks. Specifically, showing how many layers are skipped at once and how this distribution affects performance would clarify how the redundancy lies.
>
> **Response (2):** Below, we report experimental results for a fixed setting of eight pruned layers to illustrate performance variation similar to what we did in section A.10 of the appendix. The following table shows how average accuracy and cosine distance vary when fixing the number of pruned layers at 8.
>
>
> | Pruned layers   | Average acc.|   Cosine distance |
> |:----------------|-------:|--------------:|
> | 1-9             |  33.35 |          0.41 |
> | 2-10            |  34.16 |          0.4  |
> | 3-11            |  31.79 |          0.38 |
> | 4-12            |  46.3  |          0.38 |
> | 5-13            |  50.35 |          0.37 |
> | 6-14            |  52.74 |          0.37 |
> | 7-15            |  53.48 |          0.36 |
> | 8-16            |  53.67 |          0.36 |
> | 9-17            |  54.35 |          0.37 |
> | 10-18           |  52.6  |          0.38 |
> | 11-19           |  53.04 |          0.38 |
> | 12-20           |  51.38 |          0.39 |
> | 13-21           |  51.59 |          0.38 |
> | 14-22           |  54.23 |          0.37 |
> | 15-23           |  54.23 |          0.35 |
> | 16-24           |  56.15 |          0.33 |
> | 17-25           |  60.39 |          0.31 |
> | 18-26           |  58.45 |          0.28 |
> | 19-27           |  58.37 |          0.25 |
> | 20-28           |  62.04 |          0.24 |
> | 21-29           |  60.64 |          0.25 |
> | 22-30           |  61.57 |          0.25 |
> | 23-31           |  55.16 |          0.28 |
> | 24-32           |  57.65 |          0.42 |
>
>
> Additionally, the following table details the relationship between different number of pruned layers, their distributions in the original network and their corresponding average accuracy:
>
> | pruned_layers   |   Average accuracy |
> |:----------------|-------------------:|
> | 25-26           |              69.47 |
> | 24-26           |              66.47 |
> | 23-26           |              66.39 |
> | 24-28           |              64.62 |
> | 23-28           |              64.01 |
> | 22-28           |              63.34 |
> | 22-29           |              62.98 |
> | 20-28           |              60.67 |
> | 20-29           |              59.87 |
> | 19-29           |              58.97 |
> | 19-30           |              53.28 |
> | 18-30           |              51    |
> | 18-31           |              51.09 |
> | 5-19            |              46.87 |
> | 4-19            |              42.22 |
> | 3-19            |              38.66 |
> | 3-20            |              38.01 |
> | 3-21            |              38.61 |
> | 3-22            |              34.07 |
> | 3-23            |              31.78 |
> | 3-24            |              31.14 |
> | 1-23            |              29.95 |
> | 1-24            |              30.42 |
> | 1-25            |              30.22 |
>
> We hope these results bring more clarity and successfully address the reviewer's concerns. We will add this table in the Appendix.
>
> ### **Additional Parameters**
> **Comment (3):** I am curious about the size of the added linear transformations used for error compensation. It would be useful to clarify whether the number of additional parameters introduced by these layers has a significant impact on overall performance or model size. A discussion or ablation on this aspect could enhance the understanding of the method’s trade-offs.
>
> **Response (3):** We thank the reviewer for his comment. The outputs of the MLP in the $i$-th block and of the $(i+n+1)$-th block, both have shape $B \times L \times M$, where $B$ is the batch size, $L$ the sequence length, and $M$ the hidden dimension. The linear transformation that maps the MLP output of block $i$ to the output of block $i+n+1$ is therefore an $M \times M$ weight matrix. As shown in Section 2.2, this transformation **can be fused into the MLP’s down‑projection layer**, since they are two consecutive linear operations. Based on this observation **our method introduces no additional parameters**. In the revised version of the paper we will try to make this aspect clearer, so that to avoid any confusion to the readers.

---

> > ### Comment · Reviewer_PYR3 · 2025-08-02
> >
> > Thanks for the responses from the authors. I will keep my positive rating.

---

> > > ### Author Response · Authors · 2025-08-04
> > >
> > > We thank the reviewer once again for his very positive evaluation of our work.

---

### Official Review · Reviewer_ME16 · 2025-06-30

**Clarity:** 3
**Significance:** 3
**Originality:** 2
**Rating:** 4
**Confidence:** 3

**Summary:**

This paper introduces ReplaceMe - a depth pruning method that prunes transformer blocks from the base models. Pruned layers are determined by the distance between the activation outputs of different transformer blocks. In particular, they select a predefined number of layers to be pruned and seek for cut block index that minimises the distance between hidden states before and after the cut. To compensate for pruning effects, they utilise a small calibration dataset to learn a linear transformation that bridges the gap between activations before and after the cut. The authors propose to use L2 and cosine distance and find that cosine distance consistently produces better results. Besides, they propose regularisation into the calibration of linear transformation and also extend this method to non-consecutive replaced blocks. Experimental results validate the effectiveness and generality of ReplaceMe across various model families, including LLMs and Vision transformers.

**Questions:**

1. Could the authors provide more ablation studies with other similarity measures like CKA [1], CCA [2]?
2. The linear transformation is just like a feature alignment method, does it have the same role as adapters? Could we replace the linear transformation matrix with PEFT methods like Adapter or LoRA?
3. Which layers/blocks are often pruned? Could the authors provide more analysis on this?

[1] Kornblith et al., 2019. Similarity of Neural Network Representations Revisited.

[2] Morcos et. al., 2018. Insights on representational similarity in neural networks with canonical correlation.

**Ethical Concerns:**

["NO or VERY MINOR ethics concerns only"]

**Final Justification:**

After carefully reviewing all the authors' replies, all my concerns have been addressed. Therefore, I have increased my rating from 3 to 4 and updated my assessments.

**Limitations:**

Yes

**Quality:**

3

**Strengths And Weaknesses:**

## Strengths

1. The paper is well written and easy to follow
2. ReplaceMe shows better performance on various experimental settings from LLM families on NLP tasks to Vision tasks with transformer architectures with moderate compression ratios.
3. The authors validate their method with very detailed ablation studies
4. ReplaceMe significantly reduces the computational overhead in terms of latency and environmental impact.

## Weaknesses

1. This method lacks novelty since fundamental motivation and pruning technique (cosine distance) are proposed in previous work UIDL.
2. The effectiveness is only on a low/moderate compression ratio 25%.

---

> ### Author Rebuttal · Authors · 2025-07-30
>
> ### **Lack of Novelty**
> **Comment (1):** This method lacks novelty since fundamental motivation and pruning technique (cosine distance) are proposed in previous work UIDL.
>
> **Response (1):** We thank the reviewer for his comment but we respectfully disagree with his assessment. The problem of model compression is a very important one with numerous practical applications and has gathered a wide research attention since several years ago. There are many different strategies that have been proposed including pruning methods (structured, unstructured, depth-pruning), and thus we don't believe that additional motivation for investigating this problem is needed. We agree with the reviewer that the cosine distance that we use for identifying the blocks to be removed was previously expored in UIDL and ShortGPT, but we never claimed it to be part of our contribution. Our main contribution is the replacement of a sequence of layers with a linear transformation and the computation of the specific form of this transformation using calibration data. More importantly, our approach achieves model compression without a significant drop in performance while requiring a fraction of the computational cost of competing strategies, as it eliminates the need for the expensive healing process present in prior methods. Furthermore, we have conducted a detailed analysis of linear operators that feature specific properties and assessed their impact on the approximation accuracy of the original model. Finally we have performed several comparisons against competitive state-of-the-art methods both in NLP and Vision tasks and we have shown that our approach compares favorably in all cases.
> ### **Higher Compression**
> **Comment (2):** The effectiveness is only on a low/moderate compression ratio 25%.
>
> **Response (2):** The reviewer is correct that our main focus was on 25% compression ratio. The main reason for this is that with higher compression rates the compressed LLM models' performance typically drops below ~90% of the original performance. In such cases the resulting compressed model doesn't have much of practical value. Moreover, we believe that 25% is a significant pruning ratio, especially when we consider the modern LLMs who employ billions of parameters. In these  cases the number of removed parameters is in the order of hundred of millions or even billions. Nevertheless, we agree with the reviewer that evaluating a wider range of compression ratios can still be insightful. In the revised version of the paper, we will include comparisons with UIDL without healing at compression ratios up to 50%. These results are also provided below, showing the superiority of our approach even for larger compression ratios.
>
> | Model | Compression Ratio | Layers Removed | Perplexity (WikiText) ↓| Average Accuracy ↑|
> |-------|------------------|---------------|-----------------------|------------------|
> | UIDL  | 12.5%| 24-28| 21.51| 59.9|
> | Ours  | 12.5%| 24-28| **14.31**| **64.6**|
> | UIDL  | 25%| 20-28| 112.8| 50.4|
> | Ours  | 25%| 20-28| **24.8**| **60.7**|
> | UIDL  | 37.5%| 18-30| 58308| 41|
> | Ours  | 37.5%| 18-30| **192.18**| **51**|
> | UIDL  | 50%| 3-19| 140114| 30.4|
> | Ours  | 50%| 3-19| **5932**| **38.6**|
>
> In this Table we report the average performace accuracy for (Race, Winogrande, PIQA, BoolQ, OpenBookQA, and SciQ benchmarks), and the perplexity on wikitext. for Llama 3 8B instruct model.
>
> ### **CKA, CCA Ablations**
> **Comment (3):** Could the authors provide more ablation studies with other similarity measures like CKA, CCA?**
>
> **Response (3):** We agree with the reviewer that the choice of distance metric is important, and we thank him for this valuable suggestion. In response, we have conducted additional experiments using CCA and CKA distances, as presented below. These are similar to those reported in Section A.10, where we record the model's average accuracy when we prune a fixed number of layers (eight in this case) but at diffferent locations of the original model, Llama 3 8B Instruct. The last three columns of the table contain the respective values for the cosine, CCA and CKA distances.
>
>
> | Pruned layers   |   Average accuracy |   Cosine distance |   1 - CCA |   1 - CKA |
> |:----------------|-------------------:|------------------:|----------:|----------:|
> |1-9 | 33.35 |0.41 |    0.4382 |    0.6627|
> |2-10 |  34.16 |  0.4  |    0.4336 |    0.4389|
> |3-11 |  31.79 |  0.38 |    0.43   |    0.2138|
> |4-12 |  46.3  |  0.38 |    0.423  |    0.1228|
> |5-13 |  50.35 |  0.37 |    0.419  |    0.1214|
> |6-14 |  52.74 |  0.37 |    0.4143 |    0.097 |
> |7-15 |  53.48 |  0.36 |    0.4105 |    0.089 |
> |8-16 |  53.67 |  0.36 |    0.4062 |    0.1152|
> |9-17 |  54.35 |  0.37 |    0.4016 |    0.1714|
> |10-18 |   52.6  |    0.38 |    0.3964 |    0.2431|
> |11-19 |   53.04 |    0.38 |    0.39   |    0.3551|
> |12-20 |   51.38 |    0.39 |    0.3833 |    0.3445|
> |13-21 |   51.59 |    0.38 |    0.3716 |    0.3341|
> |14-22 |   54.23 |    0.37 |    0.3558 |    0.3124|
> |15-23 |   54.23 |    0.35 |    0.3362 |    0.2791|
> |16-24 |   56.15 |    0.33 |    0.3173 |    0.1893|
> |17-25 |   60.39 |    0.31 |    0.2956 |    0.1221|
> |18-26 |   58.45 |    0.28 |    0.2795 |    0.0679|
> |19-27 |   58.37 |    0.25 |    0.2685 |    **0.0484**|
> |20-28 |   **62.04** |        **0.24** |    0.2643 |    0.0529|
> |21-29 |   60.64 |    0.25 |    0.2602 |    0.0643|
> |22-30 |   61.57 |    0.25 |    **0.2592** |    0.0775|
> |23-31 |   55.16 |    0.28 |    0.2697 |    0.1226|
> |24-32 |   57.65 |    0.42 |    0.2815 |    0.1014|
>
> As we can see from the results, CCA and CKA show similar patterns to cosine distance. Their values are also high in the first layers while their minima are closer to the end of the model. However, cosine distance continues to yield superior performance: the highest accuracy is observed at the lowest cosine distance, while the accuracies corresponding to the lowest CKA and CCA distances are lower. These findings provide additional evidence that cosine distance is a reliable criterion for layer removal. We will include these results in the final version of the paper.
>
>  ### **LTs VS LoRA**
> **Comment (4):** The linear transformation is just like a feature alignment method, does it have the same role as adapters? Could we replace the linear transformation matrix with PEFT methods like Adapter or LoRA?
>
> **Response (4):** We thank the reviewer for raising this important point about the relationship between our linear transformation (LT) and parameter‑efficient fine‑tuning (PEFT) adapters. Whether one labels LT a “PEFT adapter” it ultimately depends on how one defines an adapter. In LLM Streamline, for example, pruned layers are replaced by a small MLP, what one would traditionally call an adapter, or even an entire Transformer block that is first initialized via an MSE loss and then fine‑tuned on next‑token prediction. Likewise, UIDL incorporates LoRA modules during its healing phase.
>
> Our LT could, in theory, be viewed through the same lens: it is a feature‑alignment layer that sits where a removed block once was. However, it differs fundamentally in both initialization and deployment. We compute the LT in closed form or numerically on a small calibration dataset, without backpropagating through the entire model. Once LT is calibrated, it is merged into the previous layer’s down‑projection and introduces no extra parameters at inference. Although we do explore a subsequent healing step (see Section A.14).
> Perhaps most compellingly, even without any healing, our method outperforms the more complex adapters in LLM Streamline and the LoRA‑based modules in UIDL, as shown in Table 1 in the paper. We fully acknowledge that, under very aggressive compression ratios, a simple linear map may not possess sufficient expressive power to capture all dependencies. In such extreme settings, LoRA or deep non-linear adapters might achieve a better approximation. Yet those scenarios typically incur catastrophic quality drops and require prolonged healing, which runs counter to our goal of minimal overhead and predictable, one‑shot calibration.
>
> We hope this clarifies the conceptual distinction and practical benefits of our LT approach compared to existing PEFT adapters. We would be happy to elaborate more on this if the reviewer requires more details.
>
> ### **Analysis of the Pruned Layers**
> **Comment (5):** Which layers/blocks are often pruned? Could the authors provide more analysis on this?
>
> **Response (5):** In the table included in response (3) above we report accuracy and distance metrics for all combinations of eight removed layers.
>
> In general, the most redundant layers lie in the middle and closer to the last blocks of the network, although the exact pattern varies by model. E.g., for Llama 3 8B, skipping layers 20–28 (1‑indexed) out of 32 yields the best results. We provide additional information in the following table:
>
> | Model    | Number of pruned layers | Range of pruned layers |
> | -------- | --------                | --------|
> | Llama 3 8B Instruct|2 out of 32| 24-26|
> | Llama 3 8B Instruct|4 out of 32| 24-28|
> | Llama 3 8B Instruct|8 out of 32| 20-28|
> | Falcon 2 11B|5 out of 60| 47-52|
> | Falcon 2 11B|10 out of 60| 36-46|
> | Falcon 2 11B|15 out of 60| 36-51|
> | Qwen2.5 7B Instruct|2 out of 28| 15-17|
> | Qwen2.5 7B Instruct|4 out of 28| 13-17|
> | Qwen2.5 7B Instruct|8 out of 28| 10-18|
> | Mistral 7B Instruct v0.3|2 out of 32| 22-24|
> | Mistral 7B Instruct v0.3|4 out of 32| 22-26|
> | Mistral 7B Instruct v0.3|8 out of 32| 20-28|
>
>
> It is also important to note that the optimal pruning range depends on both the architecture and the number of layers removed. Previous work has also observed that layers form natural groups [1], which we confirm using confusion matrices based on cosine similarity and CKA. While we cannot include these figures in the rebuttal due to the new NeurIPS policy, we will include them in the revised manuscript.
>
> [1] Sun et al. "Transformer Layers as Painters" (2024)

---

> > ### Comment · Reviewer_ME16 · 2025-08-03
> >
> > Dear authors,
> >
> > Thank you for your detailed response and clarification. I have carefully reviewed your replies to all reviewer comments, and I find that most of my concerns have been addressed. Therefore, I have increased the score from 3 to 4.
> >
> > Bests,

---

> > > ### Author Response · Authors · 2025-08-04
> > >
> > > We are happy that our response to the reviewer's comments has been well received and we thank the reviewer for raising his score. If there are any more clarifications or explanations needed, we are ready to provide them upon the reviewer's request.

---

### Official Review · Reviewer_wAe8 · 2025-07-02

**Clarity:** 3
**Significance:** 2
**Originality:** 2
**Rating:** 5
**Confidence:** 4

**Summary:**

This paper proposes ReplaceMe, a simple and efficient method to compress transformer-based models by replacing selected layers with linear transformations. The approach is post-hoc and training-free, requiring only a small “calibration dataset” to optimize the replacement layers via regression. The method is evaluated across a variety of language and multimodal models, demonstrating competitive performance compared to existing pruning techniques, with lower computational and environmental costs.

**Questions:**

-	Can the authors comment on using non-linear transformations instead of linear ones? Would this further improve performance?
-	How is the “environmental impact” calculated?
-	Is the learned linear transformation task-specific? That is, does each task/dataset require a new calibration and transformation? How does this compare with other pruning methods in terms of transferability?
-	The choice of calibration dataset seems to have a large impact on performance. Do the authors have any insights on the reason of this sensitivity?

**Ethical Concerns:**

["NO or VERY MINOR ethics concerns only"]

**Final Justification:**

Author addressed my concern on testing on different compression rate and adding more baselines.

**Limitations:**

yes

**Quality:**

3

**Strengths And Weaknesses:**

Strengths
-	The paper is well-written, well-structured, and easy to follow.
-	Simple yet effective idea: Replacing layers with linear transformations is a straightforward yet practical strategy that reduces computational overhead.
-	The experiments are broad, covering multiple tasks and models. The results on multi-modal models highlight the transferability of the method.
Weaknesses
-	Limited compression ratio evaluation: The experiments primarily focus on a 25% compression ratio. While this is acknowledged in the limitations section, evaluating a wider range of ratios would provide a more complete picture of the method’s robustness.
-	The primary comparison is with UIDL. Including more pruning baselines would make the comparison more convincing.
-	Missing reference: The core idea, replacing layers with linear transformations, has been explored in prior work [1, 2], although not specifically in the pruning context. These works are missing from the related work discussion.
-	The calibration dataset, a key component of the method, is only clearly explained in Section 3.3.1. Introducing this concept earlier would improve clarity and understanding.
[1] Hernandez, et al. "Linearity of relation decoding in transformer language models." ICLR 2023
[2] Wang, et al. "Lost in multilinguality: Dissecting cross-lingual factual inconsistency in transformer language models." ACL 2025

---

> ### Author Rebuttal · Authors · 2025-07-30
>
> ### **Limited Compression Ratio**
> **Comment (1)** Limited compression ratio evaluation: The experiments primarily focus on a 25% compression ratio. While this is acknowledged in the limitations section, evaluating a wider range of ratios would provide a more complete picture of the method’s robustness.
>
> **Response (1)** We appreciate the reviewer’s suggestions and we thank the reviewer for his constructive comments which will strengthen our paper.
> Compression Ratio Evaluation: While our main focus was on a 25% compression ratio (as models below ~90% original performance often cannot be successfully used in practice), we agree that evaluating a wider range of compression ratios can be insightful. We include here a comparison up to 50% and we will also include this in the final version of the paper.
>
> | Model | Compression Ratio | Layers Removed | Perplexity (WikiText) ↓| Average Accuracy ↑|
> |-------|------------------|---------------|-----------------------|------------------|
> | UIDL  | 12.5%            | 24-28         | 21.51                 | 59.9             |
> | Ours  | 12.5%            | 24-28         | **14.31**                 | **64.6**             |
> | UIDL  | 25%              | 20-28         | 112.8                 | 50.4             |
> | Ours  | 25%              | 20-28         | **24.8**                  | **60.7**             |
> | UIDL  | 37.5%            | 18-30         | 58308                 | 41               |
> | Ours  | 37.5%            | 18-30         | **192.18**                | **51**               |
> | UIDL  | 50%              | 3-19          | 140114                | 30.4             |
> | Ours  | 50%              | 3-19          | **5932**                  | **38.6**             |
>
> ### **More Pruning Baselines**
> **Comment (2)** The primary comparison is with UIDL. Including more pruning baselines would make the comparison more convincing.
>
> **Response (2)** In our initial experiments targeting LLMs, we have compared against multiple state-of-the-art model compression strategies, including SVD-LLM, LaCo, LLM-Pruner, SliceGPT, and LLM-Streamline. Please refer to tables (1, 2) in the main paper.
>
> For Vision Transformers the reviewer is correct as we indeed provide comparisons only against UIDL. The main reason for this is that almost all of the aforementioned methods have considered and provided comparisons exclusively on LLMs. In addition, our experiments on vision tasks are not by any means exhaustive, and their role is mainly to highlight that our strategy can also be successfuly deployed on vision models. Nevertheless, we agree with the reviewer that including more comparisons would be more convincing and will better highlight the benefits of our strategy. For this reason, in the updated version of our manuscript we will include additional comparisons on vision models with the LLM-streamline baseline. We include these new results below.
>
> | Model | Compression ratio | mscoco_captions (retrieval) | | cifar10 (zero-shot-classification) | | "voc2007_multilabel<br>(zero-shot-classification)" |
> |-------|-------------------|-----------------------------|---|------------------------------------|---|--------------------------------------------------|
> |       |                   | text recall@5 | vision recall@5 | acc1 | acc5 | mean_avg_p |
> | clip-vit-large-patch14 | | 0.794 | 0.611 | 0.9558 | 0.9963 | 0.7902 |
> | UIDL | 25% | 0.515 | 0.418 | 0.6933 | **0.9708** | 0.597 |
> | ReplaceMe lstsq | 25% | 0.556 | **0.471** | **0.7804** | 0.9706 | **0.688** |
> | LLMStreamline | 25% | **0.566** | 0.46 | 0.649 | 0.9515 | 0.6531 |
>
> ### **Missing Reference**
> **Comment (3)** The core idea, replacing layers with linear transformations, has been explored in prior work [1, 2], although not specifically in the pruning context. These works are missing from the related work discussion.
>
> **Response (3)** We thank the reviewer for bringing to our attention the works of ([1, 2]), which we will refer to in the related work of the revised paper.
> ### **Calibration Data**
> **Comment (4)** The calibration dataset, a key component of the method, is only clearly explained in Section 3.3.1. Introducing this concept earlier would improve clarity and understanding
>
> **Response (4)** We thank the reviewer for his very helpful comment. We agree that introducing the calibration dataset earlier would improve the clarity of the paper. For this reason, in the revised manuscript we will make explicit mention to the importance of the calibration data in Section 2 where we describe our method.
>
> ### **NonLinear Transform**
> **Comment (5)** Can the authors comment on using non-linear transformations instead of linear ones? Would this further improve performance?
>
> **Response (5)**
> * While non-linear transformations could potentially improve performance, they a) require significant fine-tuning ("healing") on large datasets and b) unlike our method, the parameters used to represent the non-linear transform cannot be directly fused with the model's existing parameters. Our key contribution is a training-free approach that achieves strong results at reasonable compression ratios (as shown in our main results tables), where we outperform methods like LLM-Streamline (that uses non-linear transformations) without requiring any healing phase.
>
> * For applications where substantial training is feasible, non-linear transformations could in principle lead to further improvements, but such a strategy would represent a different approach from our training-free method. For this reason, we have decided to focus on a practical zero-training solution, which despite its simplicity and minimal requirements can maintain a very competitive performance.
>
> ### **Enviromental Impact**
> **Comment (6)** How is the “environmental impact” calculated?
>
> **Response (6)** We calculated the environmental impact using the CodeCarbon library, which estimates energy consumption and carbon emissions during computation. We will add an explicit mention of this in the revised version of our manuscript.
>
> ### **Task Specific LTs**
> **Comment (7)** Is the learned linear transformation task-specific? That is, does each task/dataset require a new calibration and transformation? How does this compare with other pruning methods in terms of transferability?
>
> **Response (7)** We thank the reviewer for raising a very interesting point. In the context of this paper we chose open source diverse data for calibration and we didn’t explore the task-specific recovery. However, the reviewer’s comment prompted us to conduct a preliminary investigation into whether, and to what extent, performance would improve if a task-specific calibration dataset were used. To explore this question, we conducted the following experiment:
>
> 1- Use the train split of the Sciq benchmark as our calibration data.
>
> 2- Evaluate the model on the test part
>
> 3- Use a subset of Orca open-source dataset as calibration data
>
> 4- Evaluate on the same benchmarks and compare the performance for the different calibration datasets.
>
> The results are reported for Llama3 8B and we observe that indeed task-specific calibration data can further improve the performance of the compressed model. We plan to include these findings in the revised version of the paper.
>
> | Model  | Calibration Data | Compression | Sciq Accuracy |
> |--------|------------------|---------------|------------------|
> |    Llama3 8B instruct   |            -     |      -         |     0.93       |
> |    UIDL   |            Sciq- Task specific      |      25%         |     0.687       |
> |    Ours (LSTSQ)   |            Sciq- Task specific      |      25%         |     **0.89**       |
> |     Ours (LSTSQ)  |            Orca General      |        25%       |      0.858      |
>
> ### **Calibration Data Sensitivity**
> **Comment (8)** The choice of calibration dataset seems to have a large impact on performance. Do the authors have any insights on the reason of this sensitivity?
>
> **Response (8)** In Table 3 of the main manuscript, we present an evaluation of model sensitivity to the calibration data. The results indicate a dependence of model performance on data type. For example, FineWeb, which consists of raw text, leads to a degradation in performance when used for instruction-tuned models, in contrast using instruction-finetuned data such as orca or mixed-domain corpora, including a multilingual composition of 66 languages, yields more stable and consistent performance across evaluated benchmarks. On the other hand, self generated data demonstrates superior performance in terms of language modeling efficacy, as measured by perplexity. However, this advantage can be accompanied by a decline in reasoning-oriented tasks likely related to the presence of noise, factual inaccuracies, or hallucinated content within the self-generated data.
>
> In summary, our analysis reveals two principal observations regarding calibration data sensitivity:
> (1) instruction-tuned models achieve better performance when calibrated with instruction-aligned data;  (2) while self-generated data can enhance language modeling capabilities, their potential contamination with spurious or incorrect content may adversely affect performance on downstream reasoning tasks.
> These insights are solely based on our experimental results. Further investigation of this topic is very interesting and we will consider it for our future work.

---

> > ### Comment · Reviewer_wAe8 · 2025-08-04
> >
> > Thank you for the detailed response. I appreciate the detailed explanation and the additional experiments the authors provide. These help address several of my initial concerns and improve the clarity of the paper. I will raise my score accordingly.

---

> > > ### Author Response · Authors · 2025-08-05
> > >
> > > We are glad that the reviewer's comments have been successfully addressed by our response. We also would like to thank the reviewer for increasing his score.

---

### Official Review · Reviewer_MdC9 · 2025-07-02

**Clarity:** 2
**Significance:** 3
**Originality:** 3
**Rating:** 5
**Confidence:** 3

**Summary:**

This work proposes a structured pruning approach that removes transformer blocks and replaces them with a learned linear transformation. The authors argue that this method is training-free unlike other compression methods which require some fine-tuning (healing). The linear layers are estimated using a calibration dataset by matching the activations of before and after the blocks to be removed. L2 distance and cosine distance metrics are used to estimate the linear transformation from a small dataset. Experiments are comprehensive, and show that for compression ratios up to 25% the proposed method compares favourably with other methods.

**Questions:**

1. Why is this strategy well suited for transformer blocks. Why can a similar linear transformation not be used for other deep CNNs?

2. I would like the authors to speculate on the importance of choosing contiguous blocks versus optimizing the block sequence itself.

3. Clarification on performance at high compression ratio (see details under weaknesses)

**Ethical Concerns:**

["NO or VERY MINOR ethics concerns only"]

**Final Justification:**

The authors have satisfactorily addressed my concerns.

**Limitations:**

Yes

**Quality:**

3

**Strengths And Weaknesses:**

* The idea of replacing structures within a neural network with learned linear transformations is a simple but elegant idea.

* The justification of which blocks to prune based on the distance between activation maps is reasonable.

* The distance metrics used are relatively straightforward. Both L2-norm and cosine distance seem to work quite well for "reasonable" compression ratios.

* Reduction in memory achieved by using Eq. 8 is a nice addition to the contribution.

* Experiments are comprehensive; both language and vision transformers are assessed. Many recent baseline methods are also reported.

* While the compression gains are modest, the strength of the method as the authors argue lies in two factors: use of linear layers, and no fine-tuning after compression.

### Weaknesses

I have some clarifying questions.

* **Justification for removing entire blocks**: The prune and replace strategy empirically seems to work for reasonable compression ratios. How is it that these entire complex blocks so redundant? The authors point to UIDL paper as the justification. But, I would have also liked to see this argument laid out in the paper. Is it because some of the deeper layers not very useful? If so, why? Can this not be done already from the outset? Can this prune and replace strategy be used during training also? What makes this so unique to transformer blocks, and can it be applied to CNNs? If not, why?

* **Choice of blocks**: In Eq. 4 authors mention that _n_ blocks can be removed and the optimal index is obtained by minimizing the distance between transformer blocks. Why do these blocks have to be contiguous? The authors do report some of this analysis in the Appendix (Fig. 5) but I would have liked to get some justification for this in the main text too.
It would also be interesting to perform some form of a greedy search over the different transformer blocks. It would then be a sequence selection problem which would output the set of indices that need to be pruned. I can see that replacing blocks with a linear transformation is more practical, but this could also be done when disconnected blocks are removed, if the linear layer is still used just after the last removed block.
* **Difficult results tables**: I found it hard to parse the results tables. It could simply be because the authors do not elaborate the metrics being reported. For instance, what is the last column in Table 3? It just says %. I initially read it as compression ratio which you would like  to increase but then the baseline is at 100%.

* **Performance at high compression ratio**: In L189, the authors point out that at higher compression ratios their method is able to recover some performance after fine-tuning. I do not immediately see where these results are reported. Are these the results still in Fig. 3? If so, I do not understand this.
I mention this because, this would be useful to see. My hypothesis is that at higher compression ratios even large fine-tuning cannot recover the performance as linear transformations are not as expressive as something like LoRA.

* **Related work**: I find this work more closely related to the class of model compression methods that use factorization, and less to structured pruning literature. Perhaps the authors should consider describing works like [1] in their related work.

### Other Comments

* While I usually like a summary figure on page 1, I do expect it to be fully described. The figure 1 is placed even before the abstract and none of the abbreviations are introduced at this point; so the figure is misplaced. I would have preferred the method overview figure 2 in its place as it is quite self-contained.

* If reporting carbon emissions, make sure if it is CO2 or CO2 equivalent, as the latter is the more common measure.

### References

1. Ren et al. "Exploring Extreme Parameter Compression for Pre-trained Language Models" (2022)

---

> ### Author Rebuttal · Authors · 2025-07-30
>
> ### **Justification for Removing Entire Blocks:**
> **Comment (1):** How is it that these entire complex blocks so redundant? ... Is it because some of the deeper layers not very useful? If so, why?
>
> **Response (1):** The reviewer raises a valid and interesting question. While we don't have a definite answer, we hypothesize that large transformer models are often over-parameterized. This is supported by the fact that older models (like Llama 1 and 2 7B) are more compressible than newer ones (like Llama 3 8B), with the newer models being trained on significantly more data. Same findings extend to Qwen models (trained on 18T tokens), which are less compressible than Llama 3 (15T tokens). This suggests that better-trained models use their capacity more efficiently. Additionally, Huge models (e.g., Llama 3 70B) can be pruned more aggressively (up to 35%) with minimal performance loss.
>
>
> ### **Why Not Remove Redundant Blocks from the Start?**
> **Comment(2):** Can this not be done already from the outset? Can this prune and replace strategy be used during training also?
>
> **Response (2):** While our findings in this work suggest that it is possible to achieve the same performance with smaller models, trying to train from the start such a small model that matches the performance of the original larger model can be very challenging. The main reason is that the objective function being minimized during training is highly non-convex and there is always the possibility of landing to a "bad" local minima. Prior work has demonstrated that over-parameterization can help during training, makes the overall loss more amenable to optimization and improves convergence (we refer to the work on ExpandNets [Guo, Shuxuan and Alvarez, Jose M. and Salzmann, Mathieu, ExpandNets: Linear Over-parameterization to Train Compact Convolutional Networks]). Therefore, while it might be possible to design more efficient architectures upfront, current training methods seem to benefit from more parameters.
>
>
> ### **Applicability to CNNs:**
> **Comment (3):** Why is this strategy well suited for transformer blocks. Why can a similar linear transformation not be used for other deep CNNs? What makes this so unique to transformer blocks, and can it be applied to CNNs? If not, why?
>
> **Response (3):** The main reason we have focused on transformers is that they involve significanlty more parameters than CNNs, which casts pruning more impactful. While we don't believe that there is any particular reason why the same strategy couldn't be applied also to CNNs, their smaller size will most probably translate to less significant gains. However, exploring this direction could be an interesting future research direction.
> Furthermore, CNNs have different dimensions in between blocks due to the pooling operation that is typically used. In such case the learned linear transformation cannot be merged as in transformer architectures and will introduce additional parameters in the network. This will lead to smaller compression ratios given that CNNs rely on structured matrices that already use significantly less parameters than generic weight matrices.
>
> ### **Choice of Blocks**
> **Comment (4):** I would like the authors to speculate on the importance of choosing contiguous blocks versus optimizing the block sequence itself, Why do these blocks have to be contiguous? ... the linear layer is still used just after the last removed block.
>
> **Response (4):** We thank the reviewer for his suggestion. In our experiments, we have explored both consecutive and non-consecutive block removal (Appendix Table 15). While contiguous blocks are often optimal (as cosine distance tends to favor them), the best grouping can vary by model. For example, Llama 3 performs better with 2 groups of 4 continuous blocks, while Mistral works best with 4 groups of 2 consecutive blocks.
> In practice, we have observed that even when we split into groups, in most cases the chosen groups tend to also be consecutive to each other. In addition, contiguous groups allow the merged linear transformations (LTs) to be combined into a single LT, simplifying the overall process. This is also the main reason that in the main paper we report only results with a single LT. Nevertheless, our library supports both strategies, with the greedy/non-contiguous search under the $\ell_2$ loss (LSTSQ method) being fast. We will add a discussion of this in the revised version of the paper.
>
>
> ### **Difficult Results Tables**
> **Comment (5):**  what is the last column in Table 3? It just says %.
>
> **Response (5):** We thank the reviewer for highlighting this lack of clarity. The "%"  in Table 3 denotes the model’s performance relative to the original (uncompressed) model, where 100% is the baseline. Therefore, the values present in that column indicate the percentage of the original model's performance preserved after compression. We will revise the table headers and include a brief explanation in the caption to avoid any confusion to the readers.
>
>
> ### **Performance at high compression ratio**
> **Comment (6):** Clarification on performance at high compression ratio.
>
> **Response (6):** We thank the reviewer for raising this important point. For reasonable compression ratios (e.g., 25% for 7B models), our results (Appendix Table 12) show that finetuning the compressed model can recover some additional performance. In fact, an interesting finding from these results is that either training the entire model or only the Linear Transformation (LT) will lead to almost the same performance, with the latter being significantly more computationally efficient than the former. The recovery of additional performance can be attributed to the fact that our initial optimization (using $\ell_2$/cosine loss) does not directly align with the LLM training objective.
> For even higher compression ratios, we agree with the reviewer's comment and we also recommend finetuning with LoRA or full-model training, as LTs alone may lack expressiveness. While our method provides a strong starting point, the optimal approach depends on the particular model and the available computing resources and budget. Below we attach a table with the results (confirming the superiority of our approach even for larger compression ratios). We will also include a corresponding figure that depicts these results in the final version of the paper.
>
> Clarification on performance at high compression ratio
>
> | Model | Compression Ratio | Layers Removed | Perplexity (WikiText) ↓| Average Accuracy ↑|
> |-------|------------------|---------------|-----------------------|------------------|
> | UIDL  | 12.5%            | 24-28         | 21.51                 | 59.9             |
> | Ours  | 12.5%            | 24-28         | **14.31**                 | **64.6**             |
> | UIDL  | 25%              | 20-28         | 112.8                 | 50.4             |
> | Ours  | 25%              | 20-28         | **24.8**                  | **60.7**             |
> | UIDL  | 37.5%            | 18-30         | 58308                 | 41               |
> | Ours  | 37.5%            | 18-30         | **192.18**                | **51**               |
> | UIDL  | 50%              | 3-19          | 140114                | 30.4             |
> | Ours  | 50%              | 3-19          | **5932**                  | **38.6**             |
>
> In this Table (1) we report the average performace accuracy for (Race, Winogrande, PIQA, BoolQ, OpenBookQA, and SciQ benchmarks), and the perplexity on wikitext. for Llama 3 8B instruct model.
>
>
> ### **Related Work**
> **Comment (7):** Perhaps the authors should consider describing works like [1] in their related work
>
> **Response (7):** We thank the reviewer for bringing to our attention this work. The method in ([1]) is indeed relevant, as it shares similarities with SVD-LLM method,  and we will expand our related work section to include it.
>
>
> ### **Clarification (1)**
>  **Comment (8):** While I usually like a summary figure on page 1, I do expect it to be fully described. The figure 1 is placed even before the abstract and none of the abbreviations are introduced at this point; so the figure is misplaced. I would have preferred the method overview figure 2 in its place as it is quite self-contained
>
> **Response (8):** We thank the reviewer for his suggestion. We agree with his assessment and we will follow his advice to move Figure 2 in the position of Figure 1.
>
>
> ### **Clarification (2)**
> **Comment (9):** reporting carbon emissions, make sure if it is CO2 or CO2 equivalent, as the latter is the more common measure
>
> **Response (9):** We thank the reviewer for his comment. We used the publicly available tool codecarbon from github which reports emissions in CO₂ equivalent (CO₂eq), not just pure CO₂. We will make this point clear in the revised version of the manuscript.

---

> ### Comment · Reviewer_MdC9 · 2025-08-04
>
> Authors have satisfactorily addressed my concerns. I will increase my score from 4 to 5.

---

> > ### Author Response · Authors · 2025-08-04
> >
> > We thank the reviewer for his positive feedback on our rebuttal and for increasing his score.

---

### Note · Authors · 2025-08-14

## **Summary**
We would like to thank the reviewers for their positive feedbacks and the time invested in evaluating our work. In this paper we propose a novel method for model compression in transformer architectures by replacing a set of transformer blocks with linear transformations. Proposed compression approach not only preserves over 90% of the original model’s performance, but also maintains compatibility with the original architecture by fusing the linear transformation with the remaining layers (no additional parameters to save, no architectural modifications). Our approach, although conceptually simple, demonstrates strong empirical effectiveness without requiring any post-compression fine-tuning on training dataset or recovery procedures.

One of the key concerns raised by the reviewers, is the performance of our method under higher compression ratios. In rebuttal, we provide an extended evaluation including a comparative analysis with UIDL at compression ratios up to 50% of the original model size. Additionally, we present a detailed analysis of the criteria used for selecting blocks to be replaced, and conduct further experiments investigating the influence of calibration data—specifically comparing task-specific versus general (non-task-specific) datasets—on the fidelity of the compressed model. We further perform ablation studies on various distance metrics used in transformer blocks selection as recommended by one of the reviewers.

We have also addressed the technical concerns raised by the reviewers including the potential applicability of our framework to CNNs, clarifications of implementation details, and considerations regarding non-linear transformation. Regarding novelty, one of the reviewers expressed reservations, which we have carefully addressed by more clearly highlighting the unique contributions of our approach, particularly in estimating the linear transformations with a closed form-solution based on l2 objective and numerical one based on cosine similarity objective. Moreover, we provided analysis for different structured LTs, an ablation about the distance metrics and the calibration data in use. We commit to incorporating all suggested revisions and enhancements into the final version of the manuscript upon acceptance.

---

### Decision · Program_Chairs · 2025-09-17

**Decision:**

Accept (poster)

**Comment:**

This work introduces a compression method for transformer-based models that replaces selected layers with learned linear transformations. Their approach requires only a small calibration data set compared to some compression based schemes that require retraining the entire network.

Strengths:  the reviewers felt that experiments were comprehensive and that the approach yields significant performance improvements in practice.  Many reviewers also noted that, by and large, the paper was easy to read.

Weaknesses:  the reviewers identified a number of places where the exposition could be improved as well as suggested additional experimental studies that could be performed.  Most of these are quite minor, but should be addressed.

Overall:  the reviewers had a positive view on this work both before and after the discussion.  The results presented in the author feedback addressed most of the reviewers concerns.